behaviour/ecology/physiology

nutritional ecology, resource imbalance, house cricket, *Acheta domesticus*, cannibalism

**Author for correspondence:**
Yeisson Gutiérrez
e-mail: gutierrez.yeisson@gmail.com

# Diet composition and social environment determine food consumption, phenotype and fecundity in an omnivorous insect

Yeisson Gutiérrez[1], Marion Fresch[2], David Ott[1], Jens Brockmeyer[2] and Christoph Scherber[1]

[1]Institute of Landscape Ecology, University of Münster, 48149 Münster, Germany
[2]Institute for Biochemistry and Technical Biochemistry, University of Stuttgart, 70569 Stuttgart, Germany

(iD) YG, 0000-0002-0166-2933

Nutrition is the single most important factor for individual's growth and reproduction. Consequently, the inability to reach the nutritional optimum imposes severe consequences for animal fitness. Yet, under natural conditions, organisms may face a mixture of stressors that can modulate the effects of nutritional asymmetry. For instance, stressful environments caused by intense interaction with conspecifics. Here, we subjected the house cricket *Acheta domesticus* to (i) either of two types of diet that have proved to affect cricket performance and (ii) simultaneously manipulated their social environment throughout their complete life cycle. We aimed to track sex-specific consequences for multiple traits during insect development throughout all life stages. Both factors affected critical life-history traits with potential population-level consequences: diet composition induced strong effects on insect development time, lifespan and fitness, while the social environment affected the number of nymphs that completed development, food consumption and whole-body lipid content. Additionally, both factors interactively determined female body mass. Our results highlight that insects may acquire and invest resources in a different manner when subjected to an intense interaction with conspecifics or when isolated. Furthermore, while only diet composition affected individual reproductive output, the social environment would determine the number of reproductive females, thus indirectly influencing population performance.

# 1. Introduction

Considered separately, nutritional imbalance alone negatively affects a plethora of biological phenomena, for instance, development and reproduction [1–4], and additionally can induce long-lasting behavioural changes [5–7]. Likewise, the social environment bears the capacity to change gene expression patterns [8] and phenotype [9–11] in non-social animals. The combined effects of population density and food availability have been addressed in several studies [12–17]. For instance, Lihoreau *et al.* [18] pointed out the pivotal role of the nutritional status of individuals for social interactions. Yet, no studies so far have addressed to which extent the composition of available food modulates insect performance depending on the social environment.

Under natural conditions, most organisms consume a mixture of foods that vary vastly in their macronutrient composition (e.g. proteins and carbohydrates) to reach a nutritional optimum [19]. However, some scenarios, such as environments strongly transformed by intensified agriculture [20,21] or population outbreaks [22,23], may narrow the range of available nutritional sources. The resulting nutritional imbalance profoundly affects life-history traits in animals [24–26], and the consequences of such phenomena can be tracked down to the population [22,27] and community level [28]. When animals face nutritional imbalance without the chance to compensate for it (e.g. by consuming from different sources), flexibility in physiological mechanisms is decisive to survive, grow and eventually reach maturity [29,30]. In most cases, such flexibility manifests at the expense of reproductive traits, causing a trade-off between longevity and reproduction [31–33], juvenile survival and adult reproduction, or between number and size of offspring [34].

However, the nutritional mismatch (between demand and availability) is generally experienced in concert with a complex combination of environmental and social cues, which may interact to elicit different responses at the individual level [35,36]. For instance, gregarious animals living at high densities encounter an implicit trade-off as a consequence of living in groups. On the one hand, it is an effective mechanism used by many species to reduce predation risk [37,38], and on the other hand, constant interaction with conspecifics imposes high competition for space (e.g. shelter), resources and mating partners [17,22,39], and increases the risk of occurrence of cannibalism in some species [40–42]. While the effects induced by the social environment are well recognized and studied in social organisms [43,44], and in some species exhibiting parental care [45], non-social organisms also experience the (positive or negative) effects of the social environment [46]. Although the concept of 'social environment' is mainly applied in human biology [47], we find it appropriate to use it to describe the two contrasting experimental conditions that we experimentally manipulated in this study: isolation and intense interaction with conspecifics. A related concept in ecological research for such an interaction is competition, especially in regard to scramble competition [48].

In this study, we use an omnivorous insect species as a model organism, as insects are exceptional models for the study of nutritional ecology due to the ease of rearing them and their relatively short lifespan, which allows assessing effects thought their life cycle [49,50]. Besides, insects are ideal systems for the manipulation of the social environment [51], and there is a considerable body of knowledge on the optimal nutrient ratios for maximizing fitness for several insect species [52,53]. We aimed to contribute to two further knowledge gaps, namely long-term effects of nutritional imbalance and sex-specific consequences of nutritional imbalance (in both cases in combination with the manipulated social environment). It is noticeable how previous studies on insect nutritional ecology were notoriously short-term, often focusing only on late nymphal stages. Also, it has been shown that sex can determine the strategy and mechanisms by which the organisms acquire and invest their necessary nutrients [54–56]. For instance, males and females would differ in patterns of food selection due to their specific needs (reviewed in [54]). Whereas males may allocate more resources for exaggerated morphological or behavioural traits [54,55], females would invest considerably more resources into reproduction in most invertebrate and vertebrate species [24,54]. By manipulating the social environment, we tangentially address an implicit bias in former nutritional ecology experiments which, in most cases, confined the model species in solitary. Although these studies have been highly informative, these kinds of biases (i.e. short-term span, non-sex-explicit and solitary confinement) have prevented the detection of long-term effects of nutrient quality and oversimplify environmental and social components that may affect life-history traits [18].

Here, we subjected the house cricket *Acheta domesticus* (L., 1758), living in two contrasting social environments, to either of two diets differing in macronutrient constitution during their complete development. Insects were grown from the first nymphal stage until natural death and a wide range of morphological and physiological traits ranging from life-history traits up to whole-body protein

and lipid quantitation were measured with the aim to track resource allocation and investment across experimental treatments. In natural conditions, the cricket *A. domesticus* occurs in big groups in cultivated areas [57], and it has been suggested that this aggregation behaviour might be mediated by chemical cues [58]. Females are naturally larger than males and are known to choose the mating partner based on body size and song acoustic features [59,60]. Males fight over the shelter, mating partners and resources (e.g. food) [61,62].

We hypothesized that stressful conditions would negatively affect insect survival, development and reproduction and that corresponding compensatory mechanisms would be context- and sex-specific due to the different life strategies between males and females. Our specific predictions were as follows: (i) Immature crickets living in crowded conditions would experience lower survival probabilities due to severe intraspecific competition for resources (e.g. food and shelter), yet a protein-rich diet could partially alleviate this pressure. (ii) For those crickets that managed to reach adulthood, we anticipated that the type of diet they fed upon would determine their phenotype (e.g. development time, size, protein and lipid content), being benefited by a more proteinaceous diet. In this case, females would reach a bigger size in a shorter time, therefore with a higher reproductive potential; and males would benefit as well from a bigger size as this would grant them an advantage in the competition for resources (e.g. food, shelter and partners). However, we expected the social environment to modulate this response, given that a more competitive environment (i.e. intraspecific competition) would interfere with the optimal nutrient level acquisition. (iii) We predicted that reproduction would be highly determined by diet constitution as shown by previous studies [63], specifically, female crickets feeding on a protein-rich diet would have a high fecundity, at the expense of their lifespan, yet we expected the social environment to modulate this response as well as the resources gathered during the immature development, and that would be invested into reproduction in a later stage, could be compromised by intraspecific competition. Furthermore, we aimed to elucidate the role of nutrition for natural cannibalism occurring in our model organism.

# 2. Material and methods

## 2.1. Experimental design and set-up

We used the house cricket *A. domesticus* (L., 1758) as a model organism in a completely randomized two-by-two factorial design. We combined two factors considered largely influential for the development and fitness of insects: (i) diet composition, and (ii) the social environment. Diet composition was manipulated by creating two isocaloric treatments (i.e. diets) of defined chemical composition (see below) containing either 1:1 or 3:1 protein to carbohydrate ratio. Also, the social environment was manipulated in two levels by rearing insects either in solitary during their whole life cycle or grouped at high initial density (six individuals in every container, 800 inds m$^{-2}$) (figure 1). Additionally, as *A. domesticus* is known to exhibit cannibalistic behaviour [64], manipulation of social condition since egg eclosion allowed us to test for the nutritional and reproductive effects of cannibalism.

## 2.2. Diet composition

Both isocaloric diets were prepared using casein, whey protein, egg powder and sucrose (Myprotein, UK) in agar solution. Insect vitamins (Vandersant, MP Biomedicals, Germany) and a preservative (Methylparaben, Dephyte, Germany) were also added equally to the diets (for details on diets constitution and recipes, see electronic supplementary material, table S1). The levels included in this factor were chosen based on the results presented by Harrison *et al.* [63]: A protein-rich 3:1 diet (protein:carbohydrates) was hypothesized to maximize crickets' egg production and weight gain (the latter in both males and females), while lower protein contents (here considered as balanced diet—1:1) were expected to lead to a longer lifespan at the expense of reproductive performance.

## 2.3. Study species

Experimental procedures were carried out with newly hatched nymphs of the house cricket *A. domesticus*. To obtain such nymphs, young adult house crickets were purchased from a commercial supplier (Bugs International, Germany) and allowed to reproduce under laboratory conditions (30°C and 70 ± 5%

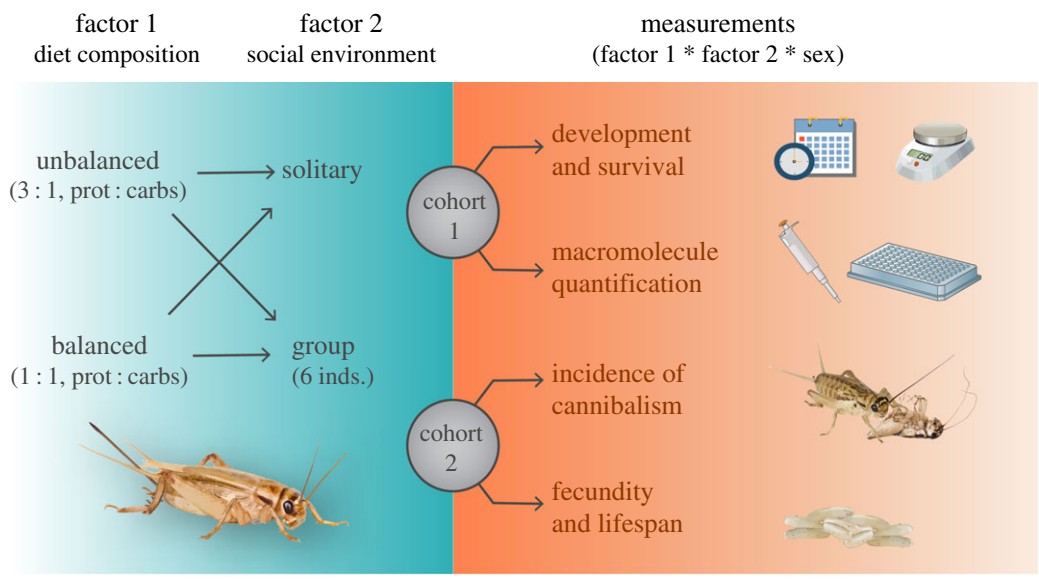

**Figure 1.** Experimental design and summary of measured response variables.

relative humidity). Food and water were offered ad libitum, and small plastic pots (20 ml) containing moist sand and covered with mosquito net were provided as an oviposition substrate. The eggs were incubated at 30°C; most of the nymphs eclosed after 12–13 days. Two different cohorts of newly hatched nymphs were subjected to identical experimental conditions to collect all intended response variables. The first cohort of nymphs was used to monitor development from the first instar until maturation and to obtain sample individuals for protein and lipid quantification. The second cohort was designated for assessing cannibalistic behaviour in grouped insects and for measuring fecundity (both in solitary and grouped insects).

Nymphs were then randomly assigned to each of the four experimental treatments (figure 1). Clear cylindrical plastic containers (i.e. experimental units—75 cm² surface area and 13 cm high, with a bottom layer of plaster of Paris) were used to enclose either grouped or single individuals. Every container was equipped with a piece of an egg carton for shelter and a 1.5 ml plastic tube plugged with cotton containing filtered water, and the food (1 ml) was provided on aluminium foil. All containers were randomized across racks in four climate cabinets (ET 650-8, Aqualytic, Germany) at 30°C and 70 ± 5% relative humidity. No light was provided inside the cabinets following the procedures of Jonsson [65]. Food and water were exchanged every week during the first month, and twice per week afterwards (when insects grew larger) to ensure ad libitum access to resources.

For the first cohort, we used 80 replicates of solitary individuals and 20 replicates of groups for each diet treatment. For the second cohort, 10 replicates for each of the four treatments.

## 2.4. Survival and development

All containers were inspected every 3–4 days until all crickets reached adulthood (ca 70 days). Dead animals (i.e. natural death) were recorded and removed if intact. Otherwise, if the loss of antennae, limbs or evident cannibalization was noted, the carcasses were left in the experimental units until consumed or dried. Time spent from egg hatching until adult emergence was recorded for every individual. Additionally, all surviving crickets were weighted at adult emergence using a precision scale (572-30, Kern, Germany).

The effects of experimental treatments on juvenile cricket survival (i.e. hazard ratio) were calculated using a mixed effects Cox model by using the coxme library [66]; diet and social environment were the fixed effects, and experimental unit code was the random factor. Development time and individual body mass were analysed with generalized linear mixed effects models (GLMMs) (function 'glmer') using the library lme4 [67], the families Poisson and Gaussian were used, respectively. Sex, diet type and social environment were the fixed effects (with three-way interactions), and an experimental unit code was the random factor in all cases. When analysing individual body mass, sex-specific models were formulated to gain a better understanding of a resulting three-way interaction.

## 2.5. Food consumption

When nymphs were approximately 50 days old (i.e. the last nymphal stage), every experimental unit received a standard portion of food ($1\ cm^3$) following the regular feeding schedule. After 4 days, the remaining food was removed, weighed and corrected for evaporation (by comparing with uneaten food in control experimental units in similar conditions without insects). For the grouped insects, consumption per individual was calculated as the total consumption divided by the actual number of surviving nymphs by day 50. Food consumption was not assessed separately for sexes within groups because it was not technically possible to track individual consumption. Data were analysed using generalized linear models (GLMs) with a Gamma (inverse link) distribution.

## 2.6. Assessment of cannibalism

In order to differentiate natural death from cannibalism in grouped insects, we conducted a thorough track of survivorship on the second cohort of insects (described above). Cricket nymphs were grouped in equal sex ratio (this could only be done from third instar onwards—where external sexual structures can be differentiated). Every dead specimen was thoroughly inspected for signs of consumption and scored as dead or cannibalized. Dead crickets were removed from the container without replacement. Cannibalism was analysed using GLMs with a quasi-binomial family and a logit link function.

## 2.7. Quantification of protein content

Five 15-day-old adult males and females from each treatment ($N = 40$) were individually homogenized in extraction buffer in a 1 : 20 ratio (w/v) (6 M urea, 1 M thiourea, 0.05 M Tris HCl, pH = 8.0) using an Ultra-Turrax T18 (IKA®-Werke GmbH & Co., Germany) at 9600 r.p.m. for 2 min. Subsequently, the samples were centrifuged at $10\,410g$ for 1 h at 4°C (Z326 K, Hermle, Germany). An aliquot of the supernatant of each sample was taken to determine protein concentration using the Bradford protein quantification assay [68]. In brief, the protein concentration of each sample is calculated using a calibration curve of a known reference sample (bovine serum albumin, Merck KGaA, Germany) using absorption spectroscopy (Spectrostar nano, BMG Labtech, Ortenberg).

Total protein content per individual was calculated as final concentration multiplied by the initial volume of extraction buffer. Relative protein content was calculated as total protein divided by the individual body mass (fresh). Both total and relative protein content were analysed using GLMs, with a Gamma distribution (inverse link) for total content and a quasi-binomial distribution (logit link) for relative content.

## 2.8. Quantification of lipid content

For this procedure, we used the same sample size and structure as for protein quantification (insects were the same age as well). The extraction of lipids was done following the method of Folch *et al.* [69] with some modifications (description of the method is provided in the electronic supplementary material). Total lipid content was calculated as lipid weight in 1 ml of homogenate (i.e. lipid concentration) multiplied by the initial volume. Relative lipid content was calculated as total lipids divided by the individual body mass fresh.

Total and relative lipid content were analysed using GLMs, with a Gamma distribution (inverse link) for total content and a quasi-binomial (logit link) distribution for relative content.

## 2.9. Reproduction and female lifespan

Ten freshly hatched adult females from all four treatments were housed individually in clean experimental units provided with food and water as in the other procedures. Each female was paired once per week (for *ca* 6 h) with a randomly selected male from a stock culture according to the method of Maklakov *et al.* [56]. Several stock populations composed of males only were maintained with the two experimental diets to ensure that females were paired with males fed with identical diets, thus avoiding uncontrolled effects. Egg production was recorded weekly (oviposition substrate similar to described above) until the natural death of all females (*ca* five months) by washing the eggs from the sand in a 300 µm sieve and counting them under a stereomicroscope.

Average weekly egg production was analysed using GLMMs (function 'glmer') with a negative binomial distribution using the library lme4 [67]. In addition, total egg production was analysed using

GLMs with a negative binomial family for heteroscedastic count data. The lifespan of females was measured from when every individual reached adulthood until its natural death, such data were analysed using a log-rank test implemented in the survival package [70]; the lifespan of males was not assessed.

## 2.10. General data analysis

For the analysis of the data collected throughout this study, we used different types of statistical models, depending on the nature of the response variable collected and the structure of the data. Mixed effects Cox models [66] were used to perform the survival analysis, including the experimental unit as a random effect (in grouped individuals). Yet, the lifespan of female *A. domesticus* was analysed using a log-rank test [71] because replicates were independent (i.e. every female was individually housed in an experimental unit). GLMs were used to model most response variables when the replicates were independent (i.e. when only one cricket per experimental unit was sampled). GLMMs [72] were used to analyse data with non-independent replicates (i.e. when several individuals from the same experimental unit had been sampled). Both types of generalized models (i.e. GLM and GLMM) were fitted using the appropriate link and variance functions (e.g. Gamma and negative binomial), which were assessed using the fitdistrplus library [73]; depending on an assessment of dispersion, we decided among (i) normal, lognormal or Gamma, (ii) Poisson or negative binomial, and (iii) binomial or quasi-binomial models [74]. Further details on the models, variables and functions are available in electronic supplementary material, table S2.

Additionally, we used structural equation models (SEM) [75] to explore life-history trade-offs among potentially correlated response variables, specifically, food consumption, development time and individual body mass (no other response variable could be included in this analysis because they were measured from only a subset of the organisms or in insects from a different batch). Data were analysed separately for each sex, and a similar conceptual model was used in both cases based on life-history theory. Average responses were calculated for each experimental unit and all variables were scaled to a numeric range of {0 : 10} [75]. Models were fitted using the lavaan library [76] with maximum-likelihood estimation with robust standard errors and a mean- and variance-adjusted test statistic (MLMVS). We tested for direct and indirect effects of the main experimental treatments on traits of adult house cricket *A. domesticus*. The exogenous variables were the experimental factors (social environment and diet composition), which were converted to binary and assumed to be fixed (without associated error terms). Development time and individual body mass were integrated into the model as response (endogenous) variables, and food consumption was used as a moderator variable. The graphical output for SEM was produced using the library semPlot [77].

All statistical analyses were performed in R 3.5.2 [78] using RStudio [79]. Type-II analysis of variance tables were used to assess the significance of terms in GLMs and GLMMs using the 'Anova' function in the car library [80]. Figures were made using the sjPlot [81] and ggplot2 [82] libraries.

# 3. Results

## 3.1. Survival of juvenile crickets and development time

The social environment was the only factor that had a noticeable effect on house cricket survival. Insects living in groups had an increased hazard of death compared to those living solitarily ($\chi^2 = 9.68$, $p = 0.002$), yet immature crickets feeding on diets of different composition did not have their survival probability affected. Living in groups increased the hazard by a factor of 0.65 (0.49–0.85, 95% CI).

The time required to reach adulthood was highly influenced by sex ($\chi^2 = 8.37$, $p = 0.004$) and diet composition ($\chi^2 = 9.28$, $p = 0.002$) and in a non-interactive way. Regardless of the diet, females completed their development faster than males, while the average for males was $64.11 \pm 1.45$ ($x \pm 95\%$ CI) days, females took $61.09 \pm 1.83$ ($x \pm 95\%$ CI) days to reach adulthood. Insects consuming high-protein diet had their development delayed, these insects took $64.7 \pm 1.75$ ($x \pm 95\%$ CI) days on average, while insects fed with the balanced diet reach adulthood in $61.27 \pm 1.45$ ($x \pm 95\%$ CI) days (figure 2a; model diagnostics available in electronic supplementary material, figure S1).

## 3.2. Individual body mass and whole-body protein and lipid content

Upon reaching adulthood, females were heavier than males in all conditions ($\chi^2 = 59.51$, $p < 0.001$), while males weighed $213.55 \pm 8.55$ ($x \pm 95\%$ CI), females achieved $282.70 \pm 17.09$ ($x \pm 95\%$ CI) mg on average.

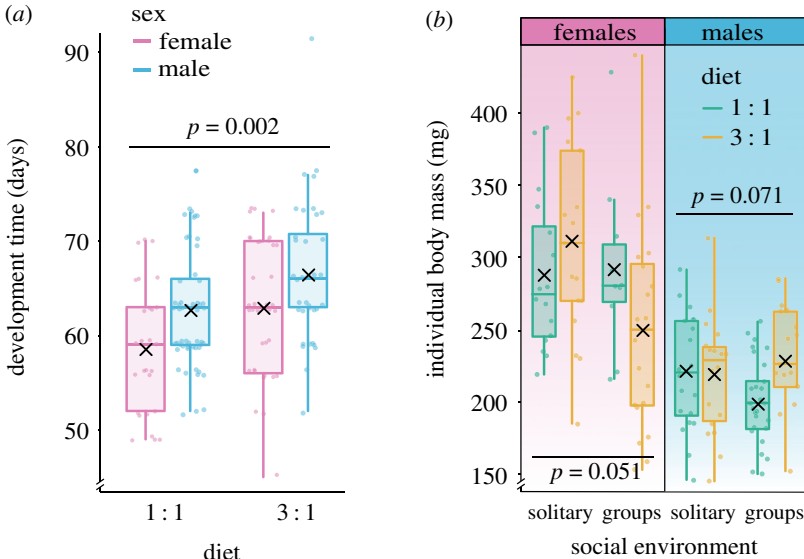

**Figure 2.** (a) Male and female house crickets A. domesticus had longer developmental time when fed with an unbalanced (protein-rich) diet. (b) Both diet composition and social environment affected the individual body mass of the house crickets in an interactive way. Among females, those living in groups and fed with an unbalanced (protein-rich) diet were the lightest, males remained unaffected by the experimental factors. Development was measured from hatching to adulthood, and individual body mass was assessed when insects reached adulthood. Development time and individual body mass were analysed with GLMMs, the families Poisson and Gaussian were used, respectively. The box plot represents the distribution of the data (minimum and maximum interquartile range) and the median of the sample, the mean is depicted by a black cross and the dots represent raw data.

Yet, diet effects depended on the social environment, and this response was different for each sex (a three-way interaction) ($\chi^2 = 7.47$, $p = 0.006$) (figure 2b; model diagnostics available in electronic supplementary material, figure S2). Separate mixed effects models for each sex revealed marginally significant effects of diet composition and social environment for females ($\chi^2 = 3.82$, $p = 0.051$), while males were not affected by any of the experimental treatments. Females living in groups and fed with the unbalanced (protein-rich) diet were the lightest with $249.95 \pm 33.67$ ($x \pm 95\%$ CI) mg, while the heaviest females were those living solitarily and fed with the protein-rich diet with $311.12 \pm 35.04$ ($x \pm 95\%$ CI) mg.

Whole-body protein content was only affected by sex ($\chi^2 = 36.41$, $p < 0.001$) and not by diet composition or social environment. Females had a higher amount of protein than males in all treatments (figure 3a; model diagnostics available in electronic supplementary material, figure S3); the content of protein in a female cricket was on average $37.04 \pm 5.87$ ($x \pm 95\%$ CI) mg and that of males was $20.56 \pm 2.18$ ($x \pm 95\%$ CI) mg. However, the relative protein content (protein/individual body mass) was similar for both sexes under all experimental conditions with $9.07 \pm 0.57$ ($x \pm 95\%$ CI) % of protein from the total fresh weight of the house crickets.

On the contrary, whole-body lipid content was affected by both sex and social environment in a two-way interaction (figure 3b; model diagnostics available in electronic supplementary material, figure S4), but diet composition did not influence this trait. Females had an overall higher lipid content than males, the content of fat in a female cricket was on average $25.49 \pm 5.33$ ($x \pm 95\%$ CI) mg and that of males was $12.54 \pm 2.17$ ($x \pm 95\%$ CI) mg. Yet, females living in groups accumulated the highest quantity of lipids in total amount ($\chi^2 = 3.94$, $p = 0.047$) with $28.55 \pm 8.50$ ($x \pm 95\%$ CI) mg, while the lowest amount of lipids was that of males living in groups with only $10.80 \pm 1.66$ ($x \pm 95\%$ CI) mg. The analysis of the relative lipid amounts only revealed differences between sexes ($\chi^2 = 5.74$, $p = 0.017$); males had $5.68 \pm 0.76$ ($x \pm 95\%$ CI) and females $7.13 \pm 0.10$ ($x \pm 95\%$ CI) % of lipids from the total fresh weight regardless of diet composition and social environment.

## 3.3. Food consumption and incidence of cannibalism

Crickets consumed a similar quantity of food regardless of diet composition, yet grouped insects had a lower food consumption on average than solitary crickets ($\chi^2 = 5.13$, $p = 0.023$). While solitary crickets (i.e. daily ingested food per capita) consumed $20.65 \pm 2.97$ ($x \pm 95\%$ CI) mg, and grouped crickets consumed

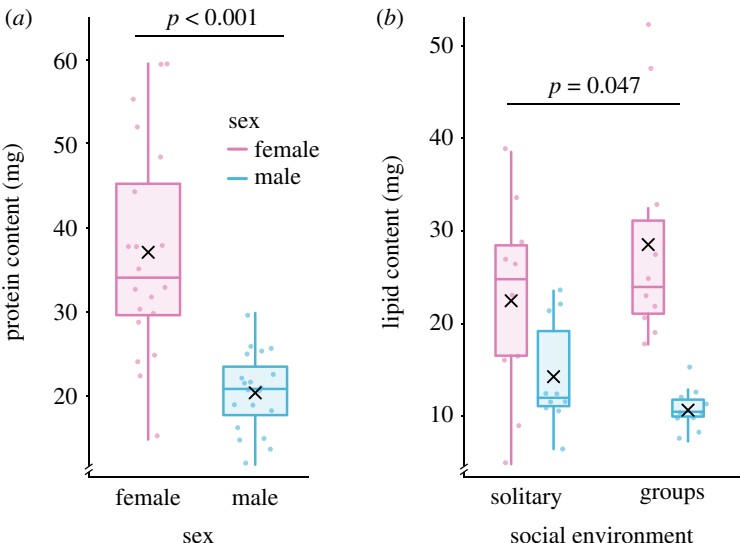

**Figure 3.** (a) Female house crickets A. domesticus had higher protein content than males, but the relative content was similar for both sexes across experimental conditions. (b) The lipid content of female house crickets was affected by the social environment; while male fat content remained unaffected by the experimental treatments (relative content exhibited a similar pattern). Both, protein and lipid content were analysed using GLMs with a Gamma distribution, using logit link in both cases. The box plot represents the distribution of the data (minimum and maximum interquartile range) and the median of the sample, the mean is depicted by a black cross and the dots represent raw data.

only $15.70 \pm 3.00$ ($x \pm 95\%$ CI) mg. Additionally, consumption differed between solitary females and males ($\chi^2 = 21.01$, $p < 0.001$); females consumed on average $26.44 \pm 5.61$ ($x \pm 95\%$ CI) mg, and males consumed $16.21 \pm 2.46$ ($x \pm 95\%$ CI) mg in a day (figure 4a; model diagnostics available in electronic supplementary material, figure S5).

The number of cannibalized house crickets was marginally affected by diet composition ($\chi^2 = 3.45$, $p = 0.063$). Insects fed with the unbalanced (protein-rich) diet had an increased rate of cannibalism ($0.37 \pm 0.16$ ($x \pm 95\%$ CI)) compared with those fed with the balanced diet ($0.22 \pm 0.10$ ($x \pm 95\%$ CI)) (figure 4b; model diagnostics available in electronic supplementary material, figure S6)

## 3.4. Reproduction and female lifespan

Females laid eggs for over seven weeks, and the dynamics of egg production were strongly affected by time ($\chi^2 = 16.108$, $p < 0.001$) and diet composition ($\chi^2 = 13.25$, $p < 0.001$) separately; on the other hand, the social environment did not affect such egg production dynamics. Females fed with the balanced (1 : 1) diet had a steady decrease in egg production during their adult life, and females fed with protein-rich (3 : 1) diet exhibited two peaks of remarkably high egg production (figure 5a; model diagnostics available in electronic supplementary material, figure S7). Regarding total egg production, females fed with the protein-rich diet had a higher overall fecundity ($424.53 \pm 229.35$ ($x \pm 95\%$ CI) eggs) than those fed with a balanced diet ($206.83 \pm 85.25$ ($x \pm 95\%$ CI) eggs) ($\chi^2 = 4.79$, $p = 0.029$; figure 5b; model diagnostics available in electronic supplementary material, figure S8).

Lifespan was severely affected only by diet composition as well ($\chi^2 = 14.40$, $p < 0.001$). Females fed with a balanced (1 : 1) diet had a longer lifespan than those fed with a protein-rich (3 : 1) diet (figure 6), while the social environment did not affect the female cricket lifespan. Feeding on a protein-rich diet increased the hazard for female crickets by a factor of 3.07 ($1.02 - 9.20$, 95% CI). The lifespan of male house crickets was not assessed.

## 3.5. Structural equation models

The SEM for female A. domesticus (figure 7a) had $\chi^2 = 1.97$ (d.f. $= 1.96$, $p = 0.364$, non-significant $p$-values indicate good fit). The full results of the model are presented in electronic supplementary material, table S3. As evidenced by the analysis previously described, the social environment affected the food consumed (figure 4a) and the attained body mass of female house crickets (figure 2b), while the diet composition

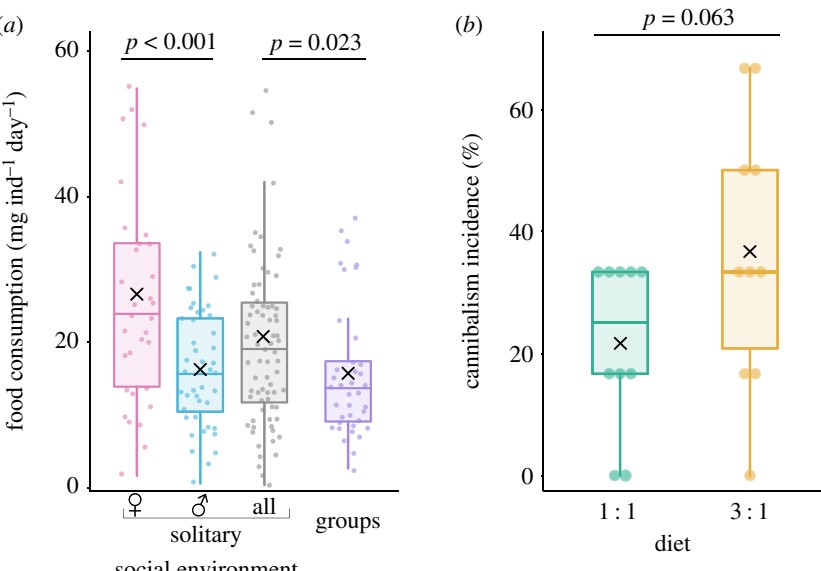

**Figure 4.** (*a*) Female house crickets *A. domesticus* consumed more food than males and grouped individuals regardless of diet composition. (*b*) The cannibalism rate was higher (marginally significant) among groups of house crickets fed with unbalanced (3 : 1) diet. Food consumption was analysed using a GLM with a Gamma distribution (inverse link), and cannibalism was analysed using a GLM with a quasi-binomial family and a logit link function. The box plot represents the distribution of the data (minimum and maximum interquartile range) and the median of the sample, the mean is depicted by a black cross and the dots represent raw data.

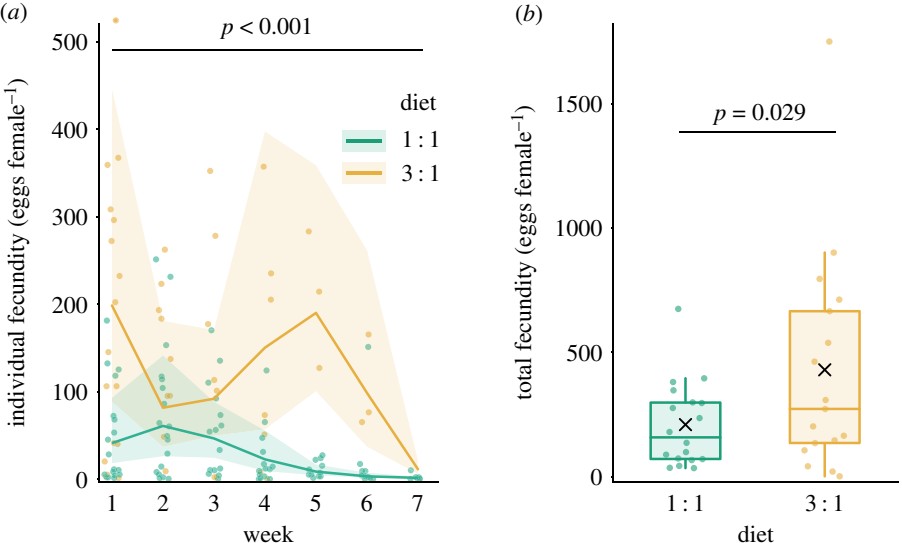

**Figure 5.** (*a*) Female house crickets *A. domesticus* fed with an unbalanced (protein-rich) diet had a higher weekly production of eggs but during a shorter span than those fed with a balanced diet. (*b*) Lifetime egg production was higher in the female house cricket fed with an unbalanced diet. Average weekly egg production was analysed using a GLMM, and total egg production was analysed using a GLM, both with a negative binomial family. The line (*a*) represents the sample mean and the variation is expressed as 95% CIs and the box plot (*b*) represents the distribution of the data (minimum and maximum interquartile range) and the median of the sample, the mean is depicted by a black cross. In both cases, the dots represent raw data.

affected the development time (figure 2*a*) and the body mass (figure 2*b*). Additionally, the SEM demonstrated that food consumption had a positive effect on female body mass, and a fairly low negative effect on development time, which implies that female crickets that ingested more food achieved a bigger size and reached adulthood earlier (i.e. shorter development time). From the SEM can also be inferred that there is a noticeable positive covariance between female cricket body mass and development time.

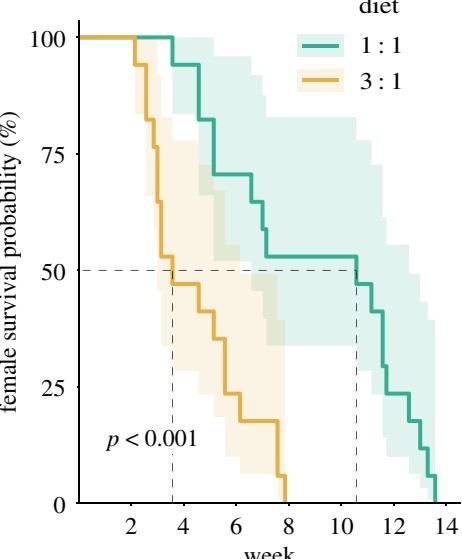

**Figure 6.** Female house crickets *A. domesticus* had a longer lifespan when feeding on a balanced (1 : 1, protein : carbohydrates) diet. Lifespan was measured from the beginning of adulthood (i.e. after final nymphal moult) and analysed using a log-rank test, the variation is expressed as 95% CIs.

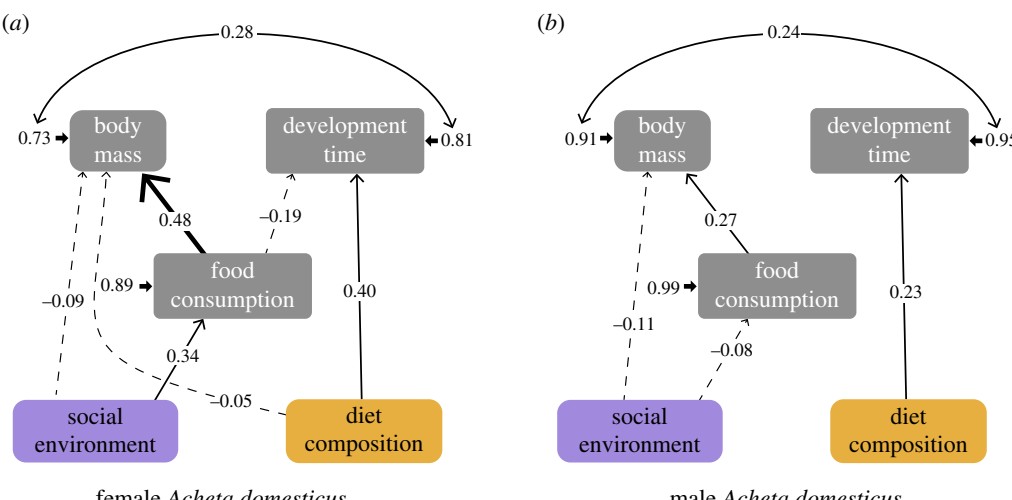

**Figure 7.** SEMs showing the direct and indirect effects of the experimental treatments, the social environment and diet composition, on the food consumption, development time and individual body mass on female (*a*) and male (*b*) *A. domesticus*. Each arrow is accompanied by standardized path coefficients. Arrow format indicates statistical significance, bold for *p* < 0.001, thin for *p* < 0.1 and dashed for non-significant relationships. Short arrows are error terms. All *p*-values for path coefficients are available in electronic supplementary material, tables S3 and S4.

The SEM for male *A. domesticus* (figure 7*b*) had $\chi^2 = 4.82$ (d.f. = 3.95, *p* = 0.301, non-significant *p*-values indicate good fit). The full results of the model are presented in electronic supplementary material, table S4. In this case, the previous analysis indicated that the diet composition affected the male cricket development time (figure 2*a*), while the social environment had an insignificant influence on male body mass (figure 2*b*) and food consumption (figure 4*a*). However, the results of the SEM show that food consumption had a positive effect on male body mass, yet food consumption was unrelated to development time. As for female crickets, male body mass exhibited a positive covariance with development time.

## 4. Discussion

The present study is, to the best of our knowledge, among the first to address the long-term consequences of the nutritional imbalance in contrasting social environments in an insect species. We addressed this

knowledge gap by designing an experiment using state-of-the-art knowledge on nutritional requirements of gryllids [1,56,63] in combination with a thorough manipulation and monitoring of the social environment of the house cricket *A. domesticus* throughout its entire life cycle. We challenged the insects with two diets of a defined constitution in the absence or presence of potential competing conspecifics, which incorporated even the risk of being cannibalized or engage in cannibalism.

We found that both treatments (i.e. diet composition and social environment) affected relevant traits in all life stages of individual insects, some of which may have far-reaching consequences at the population level due to differential survival and reproductive output. While diet composition induced expected effects on development time, lifespan and fecundity of the house cricket (as shown in previous studies [56,63]), the social environment affected the survival of immature crickets, food consumption and whole-body lipid content. Furthermore, both factors affected female body mass in an interacting manner. Additionally, we only found a marginally significant effect of diet composition on the cannibalism rate, thus suggesting that protein may not be the main driver of cannibalistic behaviour in the house cricket *A. domesticus*.

During the development of the crickets, the survival among immature individuals was decreased due to intense interaction with conspecifics, such higher overall mortality among grouped insects could be attributable to scramble competition phenomena and not to direct resource limitation as we provided unlimited access to food and water. This result contradicts the study by McFarlane [83] who proposed that the social environment would not affect cricket survival probability. Here, we suggest that the higher mortality among grouped immature crickets may have been caused by several entangled factors, namely natural mortality, intraspecific aggression due to disputes over food [61] and cannibalism.

The immature crickets that completed their development and reached adulthood achieved different sizes (i.e. individual body mass) across treatments influenced by diet composition and social environment. Although previous studies claimed that increased population density would impose a permanent negative effect on the size of individuals [65,84], our results demonstrate that this relationship is dependent on food composition and sex. Male body mass remained unaffected by diet composition and social environment, while females achieved bigger sizes living solitarily and feeding on a protein-rich diet. The complementary analysis of the life-history trait of immature crickets revealed that food consumption had a central role in the achieved body mass of both male and female adult crickets. Although food consumption was not determined by diet composition, the more food the crickets ate, the bigger body mass they achieved. Furthermore, this body size exhibited a positive covariance with the development time; which means that individuals that achieved a bigger size had a longer development time.

As discussed above, diet composition had a noticeable influence on the development time of house crickets. Even though female *A. domesticus* exhibited a remarkable development strategy by reaching maturity and gaining weight faster than males, as also shown by previous studies [85], this strategy would make females particularly susceptible to nutrition-related stress (e.g. availability and composition) as they would require to consume higher quantities of food, compared to males, to fuel such growth rates [86]. Yet, in our study, both sexes had their development significantly delayed when feeding on a protein-rich diet, which is a common response when crickets are subjected to dietary stress [87,88]. Although protein plays essential roles in organism physiology [89,90], previous studies have shown that excess dietary protein can be toxic [91,92]. Albeit diet composition did not affect immature cricket mortality, *A. domesticus* may have delayed their development as a compensatory mechanism for excess protein as both sexes, surprisingly, had similar relative body protein content across experimental conditions. It is known that *A. domesticus* is a non-specialized omnivore; in natural conditions, this cricket has been recorded attacking crops, vegetables and feeding on immobile stages of insects (e.g. eggs and pupae) [73]. Yet, the way they mix nutrients and their dietary preferences have only been studied in laboratory conditions [93–96].

By contrast to what was observed for whole-body protein (i.e. was not affected by the experimental factors), lipid content was responsive to the effects exerted by the social environment. While living in groups induced female crickets to accumulate more lipids, males living in the same conditions had the lowest lipid content. Besides their important role in the membrane structure and cell signalling, lipids are an energetic reservoir [97,98]. This finding suggests that female cricket living in crowded conditions may have stored the extra lipid content to compensate for their smaller size (i.e. body mass), although at first glance it may appear that more body lipid would have been allocated into higher egg production [99–101]. However, the latter was not true in our study because higher fecundity was only determined by higher dietary protein and not by the social environment. Males, on the other hand, would have benefited by increased lipid content as they would use this energy-dense resource for fuelling demanding activities such as chirping [87,102] and fighting [103]. Nonetheless, males living in groups

had the lowest lipid content. This suggests that the intensely competitive environment forced male crickets to spend significant amounts of energy to engage in energetically demanding physiological processes and behaviours that would allow them to reach adulthood. Meanwhile, their counterparts living in solitary confinement had higher lipid content when feeding in either diet.

A more protein-rich diet enhanced the reproductive output of the house cricket *A. domesticus*; this was consistent for females that developed in both social environments. This result is in line with previous findings for crickets [1,56,63,94] and other insect species [4,104]. In this sense, our study further emphasizes that at the individual level, nutrition is a factor of extreme relevance for fitness, with the capacity to buffer adverse social stressors. Yet, it is worth noticing that the social environment treatment was present only during half of the female lifespan (at least for those being reared in groups). Due to our experiment design, females were housed individually after reaching adulthood to control the frequency of mating and to be able to measure egg production accurately for every individual. Therefore, it is possible that prolonged interspecific interaction may cause a direct effect on female *A. domesticus* reproduction, but we cannot assess such an effect with the data yielded in this study.

The sexual dimorphism in nutrient acquisition and investment evidenced in this study might be a general phenomenon in most insect taxa as sexes may have different nutritional interests derived from disparate life strategies [54,55]. While females would gather resources during their development to afford future investment into reproduction [24], males would invest heavily in secondary sexual traits [56,105–107]. As whole-body protein was not linked to higher fecundity experienced by females fed with a protein-rich diet, we hypothesize that the underlying mechanism might be revealed through the analysis relative amounts of proteins via proteome analysis, which remains to be assessed in future studies.

The already recognized trade-off between reproduction and lifespan [4,56,91] was evident in female house crickets in this study. It has been hypothesized that egg production would compete with somatic functions for resources (i.e. nutrients) [108]. Nonetheless, recent studies suggest that this negative correlation is the product of the complex interaction of several factors (involving nutrition) in the regulation of signalling pathways [109,110].

Cannibalism is a recognized natural phenomenon occurring in *A. domesticus* [64]. Thus, a possible indirect (i.e. behavioural) regulation of resource acquisition could have occurred; this idea is further supported by the reduced overall food consumption in grouped insects. Additionally, we monitored cannibalism incidence in the cricket *A. domesticus*, and we found similar rates among insects feeding on either diet. It is noteworthy that consumption may have probably happened during the moulting period when nymphs are soft-bodied and unable to defend themselves [111]. Cannibalism is common in a wide range of species [112], and there are various causes that may prompt this behaviour. Feeding on conspecifics may serve as an alternative nutrient source in a low abundance of food or prey [113]. Our findings suggest that this was not the case for the protein requirement of individuals, and we exclude this as a driver of cannibalism in this species. During the cannibalism assessment, we noted that in most cases, the carcasses were only partially consumed, and priority had been given to limbs and portions of the exoskeleton. This particular interest in consuming the cuticle of conspecifics could be a homologue to the consumption of exuviae, which are composed mainly of nitrogenous compounds [114], particularly chitin and protein [115]. But this, once again, would have been expected in conditions of protein shortage [22,116]. Alternatively, cannibalism in the house cricket could have served as a mechanism to compensate for reduced food ingestion observed in grouped insects in this study. If seen from an opposite perspective, solitary insects may have had increased food ingestion to compensate for nutrients that they would normally have obtained by feeding on conspecifics. For instance, crickets could be consuming conspecifics seeking for lipids, which have high energy content [117,118], and according to our data, represent a fair amount of a cricket body. An additional, non-exclusive, motivation of cannibalistic behaviour could be the need to acquire microbial symbionts [119,120], but this remains to be studied in the cricket *A. domesticus*.

All considered, the combined effects of the social environment and diet composition on survival and fecundity, it can be recognized that both factors bear the potential to shape population dynamics. On the one hand, the social environment would regulate population size, and on the other hand, diet composition would determine the reproductive output. Although the social environment did not have a direct effect on reproduction, it may have important consequences by determining the number of nymphs that reached maturity and, therefore, the number of reproductive females, which would naturally affect the longer-term fitness of the population as a whole [121]. Furthermore, females living in crowded conditions have been shown to be capable to produce larger nymphs through a hormone-mediated (i.e. ecdysteroid) maternal effect [122] with potential consequences on future generations.

It is worth noticing that the house cricket *A. domesticus* used in this study may have undergone a process of selection due to artificial rearing conditions during several generations (this species is bred as animal feed [123], and the cohorts used in this experiment came from crickets purchased from a commercial company—see Material and methods for further details). As pointed out by Huey & Rosenzweig [124], the aforementioned likely selection process and the reductional nature of laboratory experiments would hinder the generalization of the results obtained after this kind of studies to wild populations.

While it has been proposed that nutrition can be a major ecological driver of social interactions [18], our study shows that the social environment itself can, in turn, mediate nutritional effects and determine impactful consequences for population size and reproduction. The interplay of both the quality of nutrients and the social environment affected important traits during the development and reproduction of the house cricket *A. domesticus*. Although density-dependent effects on insect development are well-studied phenomena [10,12,84,125], the results presented here demonstrate that accounting for the nutritional dimension can yield interesting and divergent results as nutrient quality would modulate the effects of the social environment on insect performance. Additionally, findings in this and previous studies [54–56] highlight the relevance of assessing sex-specific responses in studies dealing with the nutritional requirements of organisms.

Ethics. All applicable national guidelines for the care and use of animals were followed.

Data accessibility. All raw data has been made publicly available in the Dryad Digital Repository: https://doi.org/10.5061/dryad.fj6q573r2 [126].

Authors' contributions. Y.G., C.S. and D.O. designed the experiment. Y.G. and M.F. collected the data. Y.G. and C.S. analysed the data. Y.G. led the writing of the manuscript and all authors contributed to revisions and gave final approval for publication.

Competing interests. The authors declare that they have no competing interests.

Funding. Financial support to Y.G. was provided by Colciencias – Colombia (doctoral studies abroad program, 679-2014).

Acknowledgements. The authors thank Claudia Fricke for valuable comments and suggestions on experimental design, and Maren Zenker and Alexandra Serna for excellent technical assistance during the development of the experiments. Additionally, we acknowledge the constructive criticism of two anonymous reviewers who helped to improve our manuscript.

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
