## [Reviewer comments · Royal Society Open Science]

Review History

RSOS-191245.R0 (Original submission)

Review form: Reviewer 1

Is the manuscript scientifically sound in its present form?

No

Are the interpretations and conclusions justified by the results?

No

Is the language acceptable?

Yes

Do you have any ethical concerns with this paper?

No

Have you any concerns about statistical analyses in this paper?

Yes

Recommendation?

Reject

Comments to the Author(s)

See the attached file (Appendix A).

Review form: Reviewer 2

Is the manuscript scientifically sound in its present form?

No

Are the interpretations and conclusions justified by the results?

No

Is the language acceptable?

No

Do you have any ethical concerns with this paper?

No

Have you any concerns about statistical analyses in this paper?

No

Recommendation?

Major revision is needed (please make suggestions in comments)

Comments to the Author(s)

The authors test the individual and interactive effects of sex, diet nutrient (i.e. protein-to-carbohydrate) balance, and rearing density on cricket food consumption (including eating other crickets); development rate; body mass and composition at maturity; (juvenile?) survival and adult lifespan; and reproduction. The current presentation of the Results precludes the possibility of clearly stating general outcomes in terms of where there are interactive versus main effects of each tested factor on each type of fitness-relevant outcome. Therefore, it is equally challenging to highlight the broader intellectual insights gained from this study, as might be related to how these factors might exert interactive effects on population dynamics in the wild, where food quality and density almost certainly vary spatially and temporally for many organisms.

I recommend extensive revision and reanalysis to ensure the effort that went into this kind of labor-intensive experiment translates into substantive discussion of impactful, clearly communicated outcomes, starting with revision of the Results.

Restructure the Results to group food consumption and cannibalism results. How did food consumption rates compare between the two diets? This analysis is missing. Given that individual females ate more than individual males, there will be differences in variance when comparing food consumption between solitary individuals versus groups. This impacts the statistical comparison and interpretation of the group consumption data. Also to the extent possible distinguish whether the observed cannibalism reflects intraspecific predation versus scavenging. My years of cricket observations suggest the vast majority of incidents are scavenging, aside from the sensitive period right after a cricket molts.

Group the body mass and body composition results. Note that Maklakov et al. and Harrison et al. (cited by the authors) have already identified that there are distinct nutritional optima and

different nutrient preferences for male vs. female crickets, and thus the sex differences observed here are unsurprising. Also, males expend a considerable amount of energy via singing, a potential explanation for the lipid differences between solitary versus grouped males.

Be clear about distinctions between measurements of juvenile survival versus adult survival (lifespan). Also, explain why the impacts of grouping on adult female survival and reproduction were not assessed, or provide the analysis and outcomes. The absence of this aspect weakens the manuscript.

Introduction and Discussion: The current presented intellectual framework is weak. Clear hypotheses, predictions, and mechanistic explanations for outcomes are lacking, and multiple key citations are missing. Novelty alone is insufficient justification. E.g. How, specifically, do the authors expect that nutrient quality will "modulate" the effects of the social environment on insect fitness? I have listed a selection of references at the end of this review that should be incorporated, and would especially recommend consideration of the extensive literature on locusts, where there has been longstanding interest in understanding the interactive effects of diet and density on locust behavior and fitness outcomes.

Discussion point: Crickets could very well be eating other crickets for lipids as an energy source as much as for protein, given that the energy density of lipids is roughly twice as high as the energy density of protein (and the current data illustrate just how lipid-rich the crickets are!).

Minor comments:

Instead of "batch," use the term "cohort"

Page 2 lines 23-25: "largely unknown" does not do justice to the substantial body of literature assessing the physiological mechanisms that regulate nutrient preferences and how animals cope with imbalanced diets (reviewed in depth in reference [54]).

Page 2 lines 50/51: recommend changing "insects" to "animals."

Page 3 lines 28/29: Consider citing Zera and Tiebel (1988), as well as Mole and Zera (1994). References at the end.

Page 4 line 54: Add an indicator that "grouped" meant 6 crickets per container. Also provide full dimensions of the containers used. Surface area is more relevant than volume for interaction rates.

Page 5 line 37: Which generation(s) of lab-reared crickets were used for the experiments? Consider commentary in Huey and Rosenzweig (2009; especially p. 678 onward) and potential impacts of maternal effects, if the first generation was used.

Page 7 line 12: What age were the crickets when homogenized?

Page 8 line 21: I assume eggs were counted once a week?

Page 9 line 8: Replace "barely" with "marginally"

Page 11 line 21: You cannot conclude that "individual reproductive output was determined by nutrition alone" if there were no efforts to assess density-dependent effects on reproduction. Clarify this comparison in the Methods and state the outcome in the Results, if it was made.

Figure 2, part a, is not particularly informative and could be omitted. I would have expected to see a survival curve.

- Belovsky, G. E., & Slade, J. B. (1995). Dynamics of two Montana grasshopper populations: relationships among weather, food abundance and intraspecific competition. *Oecologia*, 101(3), 383–396. doi: 10.1007/BF00328826
- Hawlena, D., & Schmitz, O. (2010). Herbivore physiological response to predation risk and implications for ecosystem nutrient dynamics. *Proceedings of the National Academy of Sciences of the United States of America*, 107(35), 15503–15507. doi: 10.1073/pnas.1009300107
- Huey, R. B., & Rosenzweig, F. (2009). Laboratory evolution meets catch-22. In *Experimental Evolution: Concepts, Methods, and Applications of Selection Experiments* (pp. 671–701). Berkeley, CA: University of California Press.
- Lyn, J., Aksenov, V., LeBlanc, Z., & Rollo, C. D. (2012). Life History Features and Aging Rates: Insights from Intra-specific Patterns in the Cricket *Acheta domesticus*. *Evolutionary Biology*, 39(3), 371–387. doi: 10.1007/s11692-012-9160-0
- Lyn, J. C., Naikhwah, W., Aksenov, V., & Rollo, C. D. (2011). Influence of two methods of dietary restriction on life history features and aging of the cricket *Acheta domesticus*. *Age*, 33(4), 509–522. doi: 10.1007/s11357-010-9195-z
- McFarlane, J. E. (2011). A comparison of the growth of the house cricket (Orthoptera: Gryllidae) reared singly and in groups. *Canadian Journal of Zoology*. doi: 10.1139/z62-048
- Mole, S., & Zera, A. J. (1994). Differential resource consumption obviates a potential flight-fecundity trade-off in the sand cricket (*Gryllus firmus*). *Functional Ecology*, 8(5), 573–580. doi: 10.2307/2389917
- [54] Simpson, S. J., & Raubenheimer, D. (2012). *The Nature of Nutrition: A Unifying Framework from Animal Adaptation to Human Obesity*. Princeton, NJ: Princeton University Press.
- Visanuvimol, L., & Bertram, S. M. (2011). How Dietary Phosphorus Availability during Development Influences Condition and Life History Traits of the Cricket, *Acheta domesticas*. *Journal of Insect Science*, 11. doi: 10.1673/031.011.6301
- Weaver, D. K., & McFarlane, J. E. (1990). The effect of larval density on growth and development of *Tenebrio molitor*. *Journal of Insect Physiology*, 36(7), 531–536. doi: 10.1016/0022-1910(90)90105-O
- Zera, A. J., & Tiebel, K. C. (1988). Brachypterizing effect of group rearing, juvenile hormone-III and methoprene in the wing dimorphic cricket, *Gryllus rubens*. *Journal of Insect Physiology*, 34, 489–498.

Decision letter (RSOS-191245.R0)

22-Aug-2019

Dear Dr Gutiérrez Lopez:

Manuscript ID RSOS-191245 entitled "Diet quality and social environment determine phenotype, cannibalistic behaviour, and fitness in an omnivorous insect" which you submitted to Royal Society Open Science, has been reviewed. The comments from reviewers are included at the bottom of this letter.

In view of the criticisms of the reviewers, the manuscript has been rejected in its current form. However, a new manuscript may be submitted which takes into consideration these comments.

Please note that resubmitting your manuscript does not guarantee eventual acceptance, and that your resubmission will be subject to peer review before a decision is made.

Your resubmitted manuscript should be submitted by 19-Feb-2020. If you are unable to submit by this date please contact the Editorial Office.

on behalf of Dr Punidan Jeyasingh (Associate Editor) and Kevin Padian (Subject Editor)
openscience@royalsociety.org

Associate Editor Comments to Author (Dr Punidan Jeyasingh):

The authors did an ambitious and laborious experiment to explore how nutritional imbalance and social environment impacts the life history and performance of crickets. They manipulated nutritional balance and social environment to measure a number of response variables, which they have analyzed and interpreted. I have to say that the manuscript was quite hard to follow during pre-assessment, but the amount and scope of data, and the importance/relevance of the topic was quite compelling so I decided to send it for securing expert opinions. Both experts also seem to be quite enthusiastic about the topic and experiment, but were uniformly disappointed in the analyses and presentation. The reviewers elegantly furnish their concerns and recommendations. I felt the reviews were clear, fair, and constructive. With much gratitude to the expert reviewers, I hope these comments are useful to the authors in exhaustively revising their manuscript.

Reviewers' Comments to Author:

Reviewer: 1

Comments to the Author(s)

See the attached file.

Reviewer: 2

Comments to the Author(s)

The authors test the individual and interactive effects of sex, diet nutrient (i.e. protein-to-carbohydrate) balance, and rearing density on cricket food consumption (including eating other

crickets); development rate; body mass and composition at maturity; (juvenile?) survival and adult lifespan; and reproduction. The current presentation of the Results precludes the possibility of clearly stating general outcomes in terms of where there are interactive versus main effects of each tested factor on each type of fitness-relevant outcome. Therefore, it is equally challenging to highlight the broader intellectual insights gained from this study, as might be related to how these factors might exert interactive effects on population dynamics in the wild, where food quality and density almost certainly vary spatially and temporally for many organisms.

I recommend extensive revision and reanalysis to ensure the effort that went into this kind of labor-intensive experiment translates into substantive discussion of impactful, clearly communicated outcomes, starting with revision of the Results.

Restructure the Results to group food consumption and cannibalism results. How did food consumption rates compare between the two diets? This analysis is missing. Given that individual females ate more than individual males, there will be differences in variance when comparing food consumption between solitary individuals versus groups. This impacts the statistical comparison and interpretation of the group consumption data. Also to the extent possible distinguish whether the observed cannibalism reflects intraspecific predation versus scavenging. My years of cricket observations suggest the vast majority of incidents are scavenging, aside from the sensitive period right after a cricket molts.

Group the body mass and body composition results. Note that Maklakov et al. and Harrison et al. (cited by the authors) have already identified that there are distinct nutritional optima and different nutrient preferences for male vs. female crickets, and thus the sex differences observed here are unsurprising. Also, males expend a considerable amount of energy via singing, a potential explanation for the lipid differences between solitary versus grouped males.

Be clear about distinctions between measurements of juvenile survival versus adult survival (lifespan). Also, explain why the impacts of grouping on adult female survival and reproduction were not assessed, or provide the analysis and outcomes. The absence of this aspect weakens the manuscript.

Introduction and Discussion: The current presented intellectual framework is weak. Clear hypotheses, predictions, and mechanistic explanations for outcomes are lacking, and multiple key citations are missing. Novelty alone is insufficient justification. E.g. How, specifically, do the authors expect that nutrient quality will "modulate" the effects of the social environment on insect fitness? I have listed a selection of references at the end of this review that should be incorporated, and would especially recommend consideration of the extensive literature on locusts, where there has been longstanding interest in understanding the interactive effects of diet and density on locust behavior and fitness outcomes.

Discussion point: Crickets could very well be eating other crickets for lipids as an energy source as much as for protein, given that the energy density of lipids is roughly twice as high as the energy density of protein (and the current data illustrate just how lipid-rich the crickets are!).

Minor comments:

Instead of "batch," use the term "cohort"

Page 2 lines 23-25: "largely unknown" does not do justice to the substantial body of literature assessing the physiological mechanisms that regulate nutrient preferences and how animals cope with imbalanced diets (reviewed in depth in reference [54]).

Page 2 lines 50/51: recommend changing "insects" to "animals."

Page 3 lines 28/29: Consider citing Zera and Tiebel (1988), as well as Mole and Zera (1994). References at the end.

Page 4 line 54: Add an indicator that "grouped" meant 6 crickets per container. Also provide full dimensions of the containers used. Surface area is more relevant than volume for interaction rates.

Page 5 line 37: Which generation(s) of lab-reared crickets were used for the experiments? Consider commentary in Huey and Rosenzweig (2009; especially p. 678 onward) and potential impacts of maternal effects, if the first generation was used.

Page 7 line 12: What age were the crickets when homogenized?

Page 8 line 21: I assume eggs were counted once a week?

Page 9 line 8: Replace "barely" with "marginally"

Page 11 line 21: You cannot conclude that "individual reproductive output was determined by nutrition alone" if there were no efforts to assess density-dependent effects on reproduction. Clarify this comparison in the Methods and state the outcome in the Results, if it was made.

Figure 2, part a, is not particularly informative and could be omitted. I would have expected to see a survival curve.

Belovsky, G. E., & Slade, J. B. (1995). Dynamics of two Montana grasshopper populations: relationships among weather, food abundance and intraspecific competition. *Oecologia*, 101(3), 383–396. doi: 10.1007/BF00328826

Hawlena, D., & Schmitz, O. (2010). Herbivore physiological response to predation risk and implications for ecosystem nutrient dynamics. *Proceedings of the National Academy of Sciences of the United States of America*, 107(35), 15503–15507. doi: 10.1073/pnas.1009300107

Huey, R. B., & Rosenzweig, F. (2009). Laboratory evolution meets catch-22. In *Experimental Evolution: Concepts, Methods, and Applications of Selection Experiments* (pp. 671–701). Berkeley, CA: University of California Press.

Lyn, J., Aksenov, V., LeBlanc, Z., & Rollo, C. D. (2012). Life History Features and Aging Rates: Insights from Intra-specific Patterns in the Cricket *Acheta domesticus*. *Evolutionary Biology*, 39(3), 371–387. doi: 10.1007/s11692-012-9160-0

Lyn, J. C., Naikhwah, W., Aksenov, V., & Rollo, C. D. (2011). Influence of two methods of dietary restriction on life history features and aging of the cricket *Acheta domesticus*. *Age*, 33(4), 509–522. doi: 10.1007/s11357-010-9195-z

McFarlane, J. E. (2011). A comparison of the growth of the house cricket (Orthoptera: Gryllidae) reared singly and in groups. *Canadian Journal of Zoology*. doi: 10.1139/z62-048

Mole, S., & Zera, A. J. (1994). Differential resource consumption obviates a potential flight-fecundity trade-off in the sand cricket (*Gryllus firmus*). *Functional Ecology*, 8(5), 573–580. doi: 10.2307/2389917

[54] Simpson, S. J., & Raubenheimer, D. (2012). *The Nature of Nutrition: A Unifying Framework from Animal Adaptation to Human Obesity*. Princeton, NJ: Princeton University Press.

Visanuvimol, L., & Bertram, S. M. (2011). How Dietary Phosphorus Availability during Development Influences Condition and Life History Traits of the Cricket, *Acheta domestica*. *Journal of Insect Science*, 11. doi: 10.1673/031.011.6301

Weaver, D. K., & McFarlane, J. E. (1990). The effect of larval density on growth and development of *Tenebrio molitor*. *Journal of Insect Physiology*, 36(7), 531–536. doi: 10.1016/0022-1910(90)90105-O

Zera, A. J., & Tiebel, K. C. (1988). Brachypterizing effect of group rearing, juvenile hormone-III and methoprene in the wing dimorphic cricket, *Gryllus rubens*. *Journal of Insect Physiology*, 34, 489–498.

Author's Response to Decision Letter for (RSOS-191245.R0)

See Appendix B.

RSOS-200100.R0

Review form: Reviewer 3 (Szymon Drobniak)

Is the manuscript scientifically sound in its present form?

No

Are the interpretations and conclusions justified by the results?

Yes

Is the language acceptable?

Yes

Do you have any ethical concerns with this paper?

No

Have you any concerns about statistical analyses in this paper?

Yes

Recommendation?

Major revision is needed (please make suggestions in comments)

Comments to the Author(s)

The paper by Gutiérrez and colleagues looks at the combined effects of social environment and diet composition on life history and physiology of crickets. The paper has already gone through one round of reviews and I can say it is visible, the paper reads well and has logical structure. I especially like the Authors adherence to discussing biological effects instead of purely statistical significances – it's reassuring seeing papers like this, in the reproducibility era and in the crisis of overly (and wrongly) relied on p-values. However, I have some comments about the general presentation of results, the details of statistical analyses and the interpretation of the whole story.

Statistical analysis

What is the “experimental unit” mentioned in the model description? It’s important as this is critical to establish whether analyses were pseudo-replicated or not. Description of models should be possibly unambiguous. Following comment also makes it very unclear – if females were individually housed in experimental units – they still are not independent (unless the definition of experimental units clears this up for the reader).

“Integrated by independent replicates” – what does this mean?

Which modelling function (lm? lmer? glm?) as used in R?

Also, the section should describe in more detail which model was used for which type of response variable. And how these were decided. In many cases using models such as negative binomial is an overkill, and simpler Poisson models suffice. The same with gamma-distribution models. Was the decision based only on the characteristics of the data? Or on formal distribution goodness-of-fit tests? Formally testing data for fit to a specific distribution is also not recommended, especially with sample sizes seen here – such tests are overly conservative, and not very informative with smaller sample sizes.

[reading on, I see descriptions of models in respective response variables; this is unnecessarily redundant and repetitive; most of the “General data analysis” section could be removed as it is again repeated in subsequent sections]

Why food consumption was analyzed using a gamma-distributed model? I can hardly find a rational justification – a quite likely simple box-cox transformation would render the data manageable by a Gaussian distribution model.

Why introduce yet another distribution for egg counts? Log-transformation should work just fine, or (even though it has its limitations) a quasi-Poisson model, or even better – an additive overdispersion model in MCMCglmm. In cases where you claim overdispersion was tackled using quasi-Poisson – do you have indications the data IS overdispersed?

In general, I would encourage providing 95% CIs for all estimates, and not just for some (e.g. for difference in survivorship between groups etc). P values are misleading (as authors correctly argue in the response to reviewers) but this is exactly why CIs are much better in describing biological reality. It may also be using SEs but please be consistent (currently sometimes you use CIs, sometimes SE). Of course, in general confidence intervals are much more informative.

As for figures: they are much better, but still miss description – e.g. what do box plots describe? Saying “the distribution” is not enough. Are these min-max-IQR values? Are the bands on the hazard plot CIs or SEs? Also – be consistent. You report sometimes values >0.05 , and sometimes you just state n.s. I’m not a fan of p-values, but go for all or none – either everything that is >0.05 is n.s. – or report all p values (of which I’m a bigger fan actually).

Discussion

In the discussion you mention that crickets were cannibalizing only external/exoskeletal parts of other crickets (which are made of mostly indigestible chitin – is that right?). This sounds almost as if they were not really eating them but only exhibiting biting/fake eating behavior that is often seen as a response to stress in animals. In that case cannibalism would not be related to food composition and could be just reaction to stress. Are there any studies of other stress sources eliciting similar response? This warrants discussion as it’s an important component of your results.

In the discussion you also lend considerable space to the toxic effects of protein excess. However, your paper misses an important detail: what is the composition of diets of wild-type crickets? How different the two treatments you had were from natural conditions?

In general I like the discussion, it can be seen that it was worked on on structured in a better way.

Lesser points:

(Please use continuous numbering of lines in the future...)

(suggested changes in caps)

Page 2

L10: IN some scenarios

L11: such as ENVIRONMENTS STRONGLY TRANSFORMED BY ...

L13: PHENOMENA; EVEN not necessary

L16: I would remove PLASTICTY; plasticity is a tricky word and usually not equal to "flexible physiology" which may result from simple redundancy/buffering mechanisms.

L17: materializes - weird wording, maybe "is reflected in" or "manifests"?

L21: living AT high DENSITIES

L29: upper case "Isolation" not necessary

L32: what makes insects so exceptional? Others animals also eat, and often in much more interesting/biologically complex ways...

Page 3

L31 (and other throughout the text): Don't be afraid to use active voice in scientific papers. Why a long, convoluted sentence that something "was combined" - if you can say "We have ..."?

L31-32: diet quality was manipulated ... through ... diets? Rewrite, e.g. Diet was manipulated by CREATING two isocaloric treatments - or sth similar.

L34: solitarily? Perhaps - SOLITARITY?

Page 6

L15: log function - I'm assuming you are referring to log-LINK function?

L16: by non-normal errors - you refer to the original distribution of data deviating from normal? I would reword it.

Page 7

L14-16 Repetitive, described earlier in the text.

Page 10

L19 may have likely HAPPENED; by the way - do you have evidence that nymphs were more likely to be cannibalized? There is no trace of this in methods/results. If not - leave this out as this is only speculation.

L33 OF A cricket body - not "from the"

Page 12

L5 Drop "then"

I haven't made a thorough editing read of the paper but I suggest - after the whole round of revisions - an error-editing by an English editor, to get rid of possible grammar/lexical mistakes.

Review form: Reviewer 4

Is the manuscript scientifically sound in its present form?

No

Are the interpretations and conclusions justified by the results?

No

Is the language acceptable?

Yes

Do you have any ethical concerns with this paper?

No

Have you any concerns about statistical analyses in this paper?

No

Recommendation?

Major revision is needed (please make suggestions in comments)

Comments to the Author(s)

Please find my comments in attached word file (Appendix C).

Decision letter (RSOS-200100.R0)

18-Feb-2020

Dear Dr Gutiérrez Lopez,

The Subject Editor assigned to your paper ("Diet quality and social environment determine food consumption, phenotype and fitness in an omnivorous insect") has now received comments from reviewers. We would like you to revise your paper in accordance with the referee and Associate Editor suggestions which can be found below (not including confidential reports to the Editor). Please note this decision does not guarantee eventual acceptance.

Please submit a copy of your revised paper before 12-Mar-2020. Please note that the revision deadline will expire at 00.00am on this date. If we do not hear from you within this time then it will be assumed that the paper has been withdrawn. In exceptional circumstances, extensions may be possible if agreed with the Editorial Office in advance. We do not allow multiple rounds of revision so we urge you to make every effort to fully address all of the comments at this stage. If deemed necessary by the Editors, your manuscript will be sent back to one or more of the original reviewers for assessment. If the original reviewers are not available we may invite new reviewers.

When submitting your revised manuscript, you must respond to the comments made by the referees and upload a file "Response to Referees" in "Section 6 - File Upload". Please use this to

document how you have responded to each of the comments, and the adjustments you have made. In order to expedite the processing of the revised manuscript, please be as specific as possible in your response.

- Ethics statement

- Data accessibility

If you wish to submit your supporting data or code to Dryad (<http://datadryad.org/>), or modify your current submission to dryad, please use the following link:
<http://datadryad.org/submit?journalID=RSOS&manu=RSOS-200100>

- Competing interests

- Authors' contributions

- Acknowledgements

- Funding statement

on behalf of Dr Punidan Jeyasingh (Associate Editor) and Kevin Padian (Subject Editor)
 openscience@royalsociety.org

Associate Editor Comments to Author (Dr Punidan Jeyasingh):

This revised manuscript was reassessed by the original reviewers. Both reviewers felt the manuscript is much improved. Nevertheless, both of them mention several remaining issues. Again, I felt the reviews were fair and constructive. I invite the authors to make these revisions.

Editor comments:

Thank you for your revisions. As you will see, the reviewers appreciated your improvements but still have concerns. Please be aware that we cannot send your manuscript for review again. If our AE determines that you have not fully addressed all of the remaining concerns in your next version, we will not be able to consider it. Best wishes for your reworking.

Reviewer comments to Author:

Reviewer: 3

Comments to the Author(s)

The paper by Gutiérrez and colleagues looks at the combined effects of social environment and diet composition on life history and physiology of crickets. The paper has already gone through one round of reviews and I can say it is visible, the paper reads well and has logical structure. I especially like the Authors adherence to discussing biological effects instead of purely statistical significances – it's reassuring seeing papers like this, in the reproducibility era and in the crisis of overly (and wrongly) relied on p-values. However, I have some comments about the general presentation of results, the details of statistical analyses and the interpretation of the whole story.

Statistical analysis

What is the “experimental unit” mentioned in the model description? It's important as this is critical to establish whether analyses were pseudo-replicated or not. Description of models should be possibly unambiguous. Following comment also makes it very unclear – if females were individually housed in experimental units – they still are not independent (unless the definition of experimental units clears this up for the reader).

“Integrated by independent replicates” – what does this mean?

Which modelling function (lm? lmer? glm?) as used in R?

Also, the section should describe in more detail which model was used for which type of response variable. And how these were decided. In many cases using models such as negative binomial is an overkill, and simpler Poisson models suffice. The same with gamma-distribution

models. Was the decision based only on the characteristics of the data? Or on formal distribution goodness-of-fit tests? Formally testing data for fit to a specific distribution is also not recommended, especially with sample sizes seen here – such tests are overly conservative, and not very informative with smaller sample sizes.

[reading on, I see descriptions of models in respective response variables; this is unnecessarily redundant and repetitive; most of the “General data analysis” section could be removed as it is again repeated in subsequent sections]

Why food consumption was analyzed using a gamma-distributed model? I can hardly find a rational justification – a quite likely simple box-cox transformation would render the data manageable by a Gaussian distribution model.

Why introduce yet another distribution for egg counts? Log-transformation should work just fine, or (even though it has its limitations) a quasi-Poisson model, or even better – an additive overdispersion model in MCMCglmm. In cases where you claim overdispersion was tackled using quasi-Poisson – do you have indications the data IS overdispersed?

In general, I would encourage providing 95% CIs for all estimates, and not just for some (e.g. for difference in survivorship between groups etc). P values are misleading (as authors correctly argue in the response to reviewers) but this is exactly why CIs are much better in describing biological reality. It may also be using SEs but please be consistent (currently sometimes you use CIs, sometimes SE). Of course, in general confidence intervals are much more informative.

As for figures: they are much better, but still miss description – e.g. what do box plots describe? Saying “the distribution” is not enough. Are these min-max-IQR values? Are the bands on the hazard plot CIs or SEs? Also – be consistent. You report sometimes values >0.05 , and sometimes you just state n.s. I’m not a fan of p-values, but go for all or none – either everything that is >0.05 is n.s. – or report all p values (of which I’m a bigger fan actually).

Discussion

In the discussion you mention that crickets were cannibalizing only external/exoskeletal parts of other crickets (which are made of mostly indigestible chitin – is that right?). This sounds almost as if they were not really eating them but only exhibiting biting/fake eating behavior that is often seen as a response to stress in animals. In that case cannibalism would not be related to food composition and could be just reaction to stress. Are there any studies of other stress sources eliciting similar response? This warrants discussion as it’s an important component of your results.

In the discussion you also lend considerable space to the toxic effects of protein excess. However, your paper misses an important detail: what is the composition of diets of wild-type crickets? How different the two treatments you had were from natural conditions?

In general I like the discussion, it can be seen that it was worked on on structured in a better way.

Lesser points:

(Please use continuous numbering of lines in the future...)

(suggested changes in caps)

Page 2

L10: IN some scenarios

L11: such as ENVIRONMENTS STRONGLY TRANSFORMED BY ...

L13: PHENOMENA; EVEN not necessary

L16: I would remove PLASTICTY; plasticity is a tricky word and usually not equal to “flexible physiology” which may result from simple redundancy/buffering mechanisms.

L17: materializes – weird wording, maybe “is reflected in” or “manifests”?

L21: living AT high DENSITIES

L29: upper case “Isolation” not necessary

L32: what makes insects so exceptional? Others animals also eat, and often in much more interesting/biologically complex ways...

Page 3

L31 (and other throughout the text): Don’t be afraid to use active voice in scientific papers. Why a long, convoluted sentence that something “was combined” – if you can say “We have ...”?

L31-32: diet quality was manipulated ... through ... diets? Rewrite, e.g. Diet was manipulated by CREATING two isocaloric treatments – or sth similar.

L34: solitarily? Perhaps – SOLITARITY?

Page 6

L15: log function – I’m assuming you are referring to log-LINK function?

L16: by non-normal errors – you refer to the original distribution of data deviating from normal? I would reword it.

Page 7

L14-16 Repetitive, described earlier in the text.

Page 10

L19 may have likely HAPPENED; by the way – do you have evidence that nymphs were more likely to be cannibalized? There is no trace of this in methods/results. If not – leave this out as this is only speculation.

L33 OF A cricket body – not “from the”

Page 12

L5 Drop “then”

I haven’t made a thorough editing read of the paper but I suggest – after the whole round of revisions – an error-editing by an English editor, to get rid of possible grammar/lexical mistakes.

Reviewer: 4

Comments to the Author(s)

Please find my comments in attached word file.

Author's Response to Decision Letter for (RSOS-200100.R0)

See Appendix D.

Decision letter (RSOS-200100.R1)

31-Mar-2020

Dear Dr Gutiérrez Lopez,

It is a pleasure to accept your manuscript entitled "Diet composition and social environment determine food consumption, phenotype and fitness in an omnivorous insect" in its current form for publication in Royal Society Open Science.

on behalf of Dr Punidan Jeyasingh (Associate Editor) and Kevin Padian (Subject Editor)
openscience@royalsociety.org

Associate Editor Comments to Author (Dr Punidan Jeyasingh):

I thank the authors for such a thorough revision. The revised manuscript is much improved, and am I happy to recommend it for publication.

Appendix A

Summary

Both (a) nutritional imbalance and (b) population density may affect life history traits, behavior and fitness of organisms. The combined effect of both factors (a + b) on an insect was studied before, however most studies considered the food quantity and not quality. Moreover, the majority of the studies concerned only part of the insect's life cycle. This study addresses the gap, as it focuses on nutritional imbalance caused by different ratio of proteins and carbohydrates available in diet, with simultaneous consideration of social environment in a long-term study covering the whole life cycle of a house cricket *Acheta domestica* (L., 1758). The Authors are to be commended for their enthusiasm, but the manuscript is unacceptable in its present form, and should be greatly improved. After reading promising title and abstract I was disappointed with ambiguous and unintelligible description of the data analyses and results. Statistically insignificant differences ($p=0.051$; $p=0.071$; $p=0.066$ and $p=0.058$) between groups were treated as significant and discussed as proven and definitive. This concern, inter alia, cannibalistic behavior, emphasized in this study as unexpected and “challenging the idea of protein-driven cannibalism” reported in previous studies. Additionally, the graphical presentation of the results is flawed and misleading. Therefore, as an example, below I compared fig. 6 b, presented in the manuscript with more informative and not misleading version of the figure, which I created based on the raw data (all raw data has been made publicly available in the Dryad repository).

b.

On the left: “Fig. 6. (b) Cannibalism rate was higher among groups of house crickets fed with unbalance (3:1) diet. Lifespan was measured from the beginning of adulthood (i.e. after final nymphal moult). Variation expressed as confidence intervals.” (Figure caption as presented in the manuscript; in this case $p=0.058$ [see page 10, lines 6 – 10], what was not shown on the fig. 6).

On the right: when raw data are shown together with median and percentiles, the described effect may be reasonably interpreted.

Based on the fig. 6, the Authors have written in the Results section of the manuscript (page 10, lines 6 - 10): “Insects fed with the unbalanced (protein-rich) diet had an almost doubled rate of cannibalism compared with those fed with the balanced diet”. When the information on skewness of the dataset is

provided (in the form of raw datapoints), together with median and percentiles, the reader might have doubts about significance of the difference between studied groups.

Details

There are too many short paragraphs through the manuscript, what makes the text hard to process. In most cases the most important statement in the paragraph is provided first without any background support to get to that point. This structure makes the text ambiguous. Instead, the argument and background for readers should be built up in layers what will make the text easier to understand.

Abstract

line 33-34 – “non-success” is a strange expression. Please consider rewriting.

Introduction

I had a challenging time following this section. Writing is chaotic, short paragraphs are not connected with each other, various ideas and thoughts appear without context, the reader has to guess what constitutes the connective tissue of this highly fragmented stream of information. I suggest thorough rethinking and rewriting of this manuscript, focusing on one or a few ideas that should be clearly presented at the beginning of the manuscript, visible through the methods presentation, reflected in results and clearly and methodically discussed to provide to the reader clear line of thoughts through the whole manuscript.

lines:

page 3 line 6 (first sentence) – citation needed

page 3 line 17 – what is “nutritional imbalance without alternatives”? Hard to understand

page 3 lines 20-21 – what are “traditional trade-offs”? Also “trade-off among nutrition and reproduction” – what do you mean? Please be specific.

page 3 lines 21-25 – is this sentence relevant to this manuscript? Why do you mention the molecular basis and physiological mechanisms related to nutrition?

page 3 lines 30-35 - are these sentences relevant? Why do you mention thermal plasticity, pesticides, social organisms and parental care?

page 3 line 45 - page 4 line 7 – this could be a nice introducing paragraph

page 3 lines 52 – 54 – remove “A more recent study by”

page 3 line 56 – remove “i.e. composition”

page 3 lines 56-58 – consider “quality of available food” instead of “quality of available nutrients”

page 4 line 7 – consider “resource quantity” instead of “resource availability (i.e. density)”

page 4 lines 8 – 17 – consider condensing this paragraph to one sentence

page 4 line 24 – I don't understand why nymphal stages are the most important parts of life cycles

page 4 lines 30 – 33 – consider elaborating more on differences between sexes.

Materials and Methods

Statistical analyses were described very briefly without giving details needed to understand what actually was computed. Did you do regression diagnostics for your models? This information should be given and details should be presented in supplement. In Results section you give always χ^2 and p values. What test was computed: likelihood – ratio chi – square test or Type II Wald chi – square test? Why F, Df and p values were not used instead?

Please explain briefly for all the statistical analyses separately why did you use particular model.

Additionally, you should clearly describe here, how do you treat the p values. You described the p values higher than 0.05 as “barely statistically significant” and you treat differences between groups with $p > 0.05$ as statistically significant. Why? ($p > 0.05$ is given four times: on page 9, lines 5-8, and 45-46, and page 10, lines 6-7).

lines:

page 4 line 60 – remove “In this study”

page 4 line 60 – page 5 line 8 – very laborious to read; please rewrite

page 6 lines 20 – 22 – “data were summarized” – please elaborate and be specific

page 6 lines 31-32 – please clarify “dead animals” (I guess – naturally dead and not cannibalized)

page 7 lines 43 – 58 – please transfer this paragraph to the supplement

Results

Results are unclear and graphics do not provide needed information. It is not known when differences between groups are statistically significant. In most cases it is written that factor studied was lower or higher in one group than in the other without giving specific values. When writing about factors like body mass, protein content number of eggs laid, etc. the Authors should give mean / median values and show how big was the difference. Based on the comparison of figures presented at the beginning of this report I also suspect that the skew of the data distribution and homogeneity / heterogeneity of variance were not taken into account when presenting the results, resulting in false statements like on page 10, lines 6-10: “Insects fed with the unbalanced (protein-rich) diet had an almost doubled rate of cannibalism compared with those fed with the balanced diet”.

Nothing more than in the text is shown on the figures. The figures should not be the exact reflection of the text but should provide additional information. They should be also self – descriptive. Here, there is no information provided on what actually is shown (what are dots and bars? differently calculated confidence intervals may be used with R car library, so which ones were used?). There is no information on the figures about statistics done and statistical differences between groups.

To allow the reader for the correct interpretation of the data presented on the figures I strongly suggest to show raw data and mean in case of normal distribution and raw data, median and percentiles in case of skewed distribution.

Discussion

I believe this section should be completely rewritten and the strength of conclusions should be reduced after reconsidering the results of the study. The title and abstract should be changed accordingly.

I am not able to propose major revision of the manuscript in the current form. I hope that my comments will help the Authors to improve the manuscript and I will be very happy to see the improved version published. As an author of several papers I know how hard is the process of publishing a manuscript. While I had to be critical in my evaluation, I keep my fingers crossed for the Authors.

Kind regards,
Reviewer

Appendix B

Response to decision letter

Dear Dr Gutiérrez Lopez:

Manuscript ID RSOS-191245 entitled "Diet quality and social environment determine phenotype, cannibalistic behaviour, and fitness in an omnivorous insect" which you submitted to Royal Society Open Science, has been reviewed. The comments from reviewers are included at the bottom of this letter.

In view of the criticisms of the reviewers, the manuscript has been rejected in its current form. However, a new manuscript may be submitted which takes into consideration these comments.

Please note that resubmitting your manuscript does not guarantee eventual acceptance, and that your resubmission will be subject to peer review before a decision is made.

Your resubmitted manuscript should be submitted by 19-Feb-2020. If you are unable to submit by this date please contact the Editorial Office.

Response: We are very thankful for the detailed and fast feedback on our manuscript. We tried our best to address all concerns from both reviewers, and we hope that after the substantial changes herein included, our manuscript can be accepted for publication in "Royal Society Open Science".

We kindly ask the Editorial Bboard to send the corrected manuscript to the same reviewers to continue with the line of thought, and also, because they proved to be keen on the topic.

Reviewers' Comments to Author:

Reviewer: 1

Both (a) nutritional imbalance and (b) population density may affect life history traits, behavior and fitness of organisms. The combined effect of both factors (a + b) on an insect was studied before, however most studies considered the food quantity and not quality. Moreover, the majority of the studies concerned only part of the insect's life cycle. This study addresses the gap, as it focuses on nutritional imbalance caused by different ratio of proteins and carbohydrates available in diet, with simultaneous consideration of social environment in a long-term study covering the whole life cycle of a house cricket *Acheta domesticus* (L., 1758). The Authors are to be commended for their enthusiasm, but the manuscript is unacceptable in its present form, and should be greatly improved. After reading promising title and abstract I was disappointed with ambiguous and unintelligible description of the data analyses and results. Statistically insignificant differences ($p=0.051$; $p=0.071$; $p=0.066$ and $p=0.058$) between groups were treated as significant and discussed as proven and definitive.

Response: We are aware of the controversy around p-values. Consequently, we have now modified the revised version of our manuscript to include as “marginally significant” those p-values ranging from 0.05 to 0.06 as long as the effect is considerable, in line with recent recommendations in ongoing debates on p-values (e.g. Amrhein *et al.* 2019; Wasserstein *et al.* 2019). Our aim was not to oversell our results, but rather not to dismiss biologically relevant findings based merely on statistical tests. Therefore, we address this issue by differentiating significant differences in the traditional sense ($P<0.05$) and “marginally significant” for p-values slightly larger than 0.05 (yet, all results with p-values >0.06 were not considered “marginally significant” as in the previous version). We hope that these corrections and clarifications now made the manuscript more straightforward for the readers.

Amrhein, V., Greenland, S., & McShane, B. (2019). Scientists rise up against statistical significance. *Nature* 567, 305-307

Wasserstein, R. L., Schirm, A. L., & Lazar, N. A. (2019). Moving to a world beyond “ $p<0.05$ ”. *The American Statistician* 73, 1-19

This concern, inter alia, cannibalistic behavior, emphasized in this study as unexpected and “challenging the idea of protein-driven cannibalism” reported in previous studies. Additionally, the graphical presentation of the results is flawed and misleading. Therefore, as an example, below

I compared fig. 6 b, presented in the manuscript with more informative and not misleading version of the figure, which I created based on the raw data (all raw data has been made publicly available in the Dryad repository).

On the left: “Fig. 6. (b) Cannibalism rate was higher among groups of house crickets fed with unbalance (3:1) diet. Lifespan was measured from the beginning of adulthood (i.e. after final nymphal moult). Variation expressed as confidence intervals.” (Figure caption as presented in the manuscript; in this case $p=0.058$ [see page 10, lines 6 – 10], what was not shown on the fig. 6).

On the right: when raw data are shown together with median and percentiles, the described effect may be reasonably interpreted. Based on the fig. 6, the Authors have written in the Results section of the manuscript (page 10, lines 6 - 10): “Insects fed with the unbalanced (protein-rich) diet had an almost doubled rate of cannibalism compared with those fed with the balanced diet”. When the information on skewness of the dataset is provided (in the form of raw datapoints), together with median and percentiles, the reader might have doubts about significance of the difference between studied groups.

Response: We thank the referee for these detailed suggestions. We have now modified the Figure (now Fig. 5b) to a box-plot and included raw data points to improve clarity.

In addition, lines 3 – 5 on page 9 were rewritten to avoid overstating the results:

“The number of cannibalized house crickets was marginally affected by diet quality ($\chi^2 = 3.57, P = 0.058$). Insects fed with the unbalanced (protein-rich) diet had an increased rate of cannibalism (0.37 ± 0.07 ($\bar{x} \pm SE$)) compared with those fed with the balanced diet (0.22 ± 0.04 ($\bar{x} \pm SE$)) (Fig. 3b, model diagnostics available in the electronic supplementary material as Fig. S6)”.

Details

There are too many short paragraphs through the manuscript, what makes the text hard to process. In most cases the most important statement in the paragraph is provided first without any background support to get to that point. This structure makes the text ambiguous. Instead, the argument and background for readers should be built up in layers what will make the text easier to understand.

Response: We thank the reviewer for pointing to this and providing guidance on how to structure a clearer manuscript. We reorganized the introduction and discussion, and reduced unnecessary subdivision of paragraphs as suggested.

Abstract

line 33-34 – “non-success” is a strange expression. Please consider rewriting.

Response: We have rewritten these lines (page 1, L 1-2) as follows “Consequently, the inability to reach the nutritional optimum...”.

Introduction

I had a challenging time following this section. Writing is chaotic, short paragraphs are not connected with each other, various ideas and thoughts appear without context, the reader has to guess what constitutes the connective tissue of this highly fragmented stream of information. I suggest thorough rethinking and rewriting of this manuscript, focusing on one or a few ideas that should be clearly presented at the beginning of the manuscript, visible through the methods presentation, reflected in results and clearly and methodically discussed to provide to the reader clear line of thoughts through the whole manuscript.

Response: We restructured and narrowed down the ideas presented in the introduction to create a more coherent and cohesive manuscript.

lines:

page 3 line 6 (first sentence) – citation needed

Response: We included the following citations for this information (page 2, L 13):

Smykal V, Raikhel AS. 2015 Nutritional control of insect reproduction. *Curr. Opin. insect Sci.* 11, 31–38.

Simpson SJ, Raubenheimer D. 2012 *The nature of nutrition: a unifying framework from animal adaptation to human obesity.* Princeton University Press.

Kolss M, Vijendravarma RK, Schwaller G, Kawecki TJ. 2009 Life-History consequences of adaptation to larval nutritional stress in *Drosophila*. *Evolution* (N. Y). 63, 2389–2401.

page 3 line 17 – what is “nutritional imbalance without alternatives”? Hard to understand

Response: We have rewritten these lines (page 2, L 14-16) as “When animals face nutritional imbalance without the chance to compensate for it (e.g. by consuming from different sources), flexibility in physiological mechanisms (i.e. plasticity) is decisive to survive, grow, and eventually reach maturity [29,30].”.

page 3 lines 20-21 – what are “traditional trade-offs”? Also “trade-off among nutrition and reproduction” – what do you mean? Please be specific.

Response: We have rewritten these lines (page 2, L 16-18) as “In most cases, such plasticity materializes at the expense of reproductive traits, causing a trade-off between longevity and reproduction [31–33].”.

page 3 lines 21-25 – is this sentence relevant to this manuscript? Why do you mention the molecular basis and physiological mechanisms related to nutrition?

Response: We agree with Reviewer 1 on that this sentence does not fit well in the manuscript. We proceeded to remove it accordingly.

page 3 lines 30-35 - are these sentences relevant? Why do you mention thermal plasticity, pesticides, social organisms and parental care?

Response: We removed the lines referring to thermal plasticity and pesticides in the context of nutrition as they did not add to the storyline of the manuscript. Yet, we kept and rearranged the sentence on parental care as this is useful to point the knowledge gap on the study of the social environment in non-social organisms (page 2, L 25-27).

page 3 line 45 - page 4 line 7 – this could be a nice introducing paragraph

Response: We have repositioned these paragraphs at the beginning of the introduction as suggested.

page 3 lines 52 – 54 – remove “A more recent study by”

Response: Removed as suggested.

page 3 line 56 – remove “i.e. composition”

Response: Removed as suggested.

page 3 lines 56-58 – consider “quality of available food” instead of “quality of available nutrients”

Response: We replaced this line following the suggestion of Reviewer 1.

page 4 line 7 – consider “resource quantity” instead of “resource availability (i.e. density)”

Response: We replaced this line following the suggestion of Reviewer 1.

page 4 lines 8 – 17 – consider condensing this paragraph to one sentence

Response: We considerably reduced the length of this paragraph, keeping only the most relevant information.

page 4 line 24 – I don’t understand why nymphal stages are the most important parts of life cycles

Response: This part was deleted to avoid confusion. The lines will remain as follows (page 2, L 36): “It is noticeable how previous studies on insect nutritional ecology were notoriously short-term, often focusing only on late nymphal stages”.

page 4 lines 30 – 33 – consider elaborating more on differences between sexes.

Response: We extended this paragraph with the examples from the cited studies. The paragraph now reads as follows (page 3, L 1-7):

“Also, it has been shown that sex can determine the strategy and mechanisms by which the organisms acquire and invest their necessary nutrients [50–52]. For instance, males and females would differ in patterns of food selection due to their specific needs (reviewed in [50]). Whereas males may allocate more resources for exaggerated morphological or behavioural traits [50,51], females would invest considerably more resources into reproduction in most invertebrate and vertebrate species [24,50]. By manipulating the social environment, we tangentially address an implicit bias in former nutritional ecology experiments which, in most cases, confined the model species in solitary.”.

Materials and Methods

Statistical analyses were described very briefly without giving details needed to understand what actually was computed. Did you do regression diagnostics for your models? This information should be given and details should be presented in supplement.

Response: We included a new paragraph (page 5, L 5-15) in the subheading “General data analysis” of the materials and methods section addressing the usage of every model or statistical analysis conducted in our study. Additionally, we incorporated regression diagnostics for the models in the electronic supplementary material.

In Results section you give always χ^2 and p values. What test was computed: likelihood – ratio chi – square test or Type II Wald chi – square test? Why F, Df and p values were not used instead?

Response:

The Chi^2 values reported here are all based on one degree of freedom as they employ type-II- Chi^2 tests on model terms. Each term is tested by adding it to a model including all others (see Fox and Weisberg 2019, page 262).

We have now explained this better in the section on “General data analysis” where we state that the significance of all GLM and GLMM models was computed using the 'Anova' function in the car library.

Reference:

Fox J, Weisberg S (2019) An R companion to applied regression. 3rd edn., SAGE Publications, Inc.

Please explain briefly for all the statistical analyses separately why did you use particular model.

Response: We included a full new paragraph in the section “General data analysis” addressing this issue (page 5, L 5-15).

Additionally, you should clearly describe here, how do you treat the p values. You described the p values higher than 0.05 as “barely statistically significant” and you treat differences between groups with $p > 0.05$ as statistically significant. Why? ($p > 0.05$ is given four times: on page 9, lines 5-8, and 45-46, and page 10, lines 6-7).

Response: We changed all sentences involving P-values in the range of 0.050 – 0.060 to “marginally significant”, and removed results with $P > 0.060$ to adopt a more homogeneous language throughout the manuscript.

lines:

page 4 line 60 – remove “In this study”

Response: Removed as suggested.

page 4 line 60 – page 5 line 8 – very laborious to read; please rewrite

Response: We rewrote the whole paragraph as follows (page 3, L 29 – page 4, L 3):

“We used the house cricket *Acheta domesticus* (L. 1758) as a model organism in a completely randomized two-by-two factorial design. Two factors considered largely influential for the development and fitness of insects were combined: 1) Diet quality (i.e. composition) and 2) the social environment. Diet quality was manipulated in two levels through the isocaloric diets of defined chemical composition (see below) containing either 1:1 or 3:1 protein to carbohydrate ratio. Also, the social environment was manipulated in two levels by rearing insects either in solitarily during their whole life cycle or grouped at high initial density (six individuals in every container, 800 inds/m²) (Fig. 1). Additionally, as *A. domesticus* is known to exhibit cannibalistic behaviour [54], manipulation of social condition since egg eclosion allowed us to test for the nutritional and reproductive effects of cannibalism”.

page 6 lines 20 – 22 – “data were summarized” – please elaborate and be specific

Response: We deleted this sentence and the corresponding reference as we did not use this package in the final statistical analysis (it was used in the initial exploration of the data but was mistakenly included in the manuscript).

page 6 lines 31-32 – please clarify “dead animals” (I guess – naturally dead and not cannibalized)

Response: We modified the sentence as follows “Dead animals (i.e. natural death) were recorded ...” (page 5, L 21-22).

page 7 lines 43 – 58 – please transfer this paragraph to the supplement

Response: Transferred to the electronic supplementary material as suggested.

Results

Results are unclear and graphics do not provide needed information. It is not known when differences between groups are statistically significant. In most cases it is written that factor studied was lower or higher in one group than in the other without giving specific values.

Response: We included p-values in all graphs to make graphs clearer for the readers.

When writing about factors like body mass, protein content number of eggs laid, etc. the Authors should give mean / median values and show how big was the difference.

Response: We included this information in the results section (pages 7-9) for nymph survival, development time, weight, food consumption, protein content, lipid content, total egg production, female lifespan and rate of cannibalism.

Based on the comparison of figures presented at the beginning of this report I also suspect that the skew of the data distribution and homogeneity / heterogeneity of variance were not taken into account when presenting the results, resulting in false statements like on page 10, lines 6-10: “Insects fed with the unbalanced (protein-rich) diet had an almost doubled rate of cannibalism compared with those fed with the balanced diet”.

Response: We did account for data distribution and heterogeneity of variance. As explained in the new paragraph included in the “general data analysis” subsection, we selected the adequate family for every GLM or GLMM model depending on data distribution that had formally been checked based on statistical properties of each variable (e.g. being count data or binomial) and using R’s `fitdistrplus` package.

Nothing more than in the text is shown on the figures. The figures should not be the exact reflection of the text but should provide additional information. They should be also self – descriptive. Here, there is no information provided on what actually is shown (what are dots and bars? Differently calculated confidence intervals may be used with R `car` library, so which ones were used?). There is no information on the figures about statistics done and statistical differences between groups.

Response: As commented above, we have included the p-values for all graphs. Additionally, we expanded the figure captions (pages 18-19) by explaining the statistical

procedures (models used for every analysis), box-plots, mean (black crosses) and points representing raw data.

To allow the reader for the correct interpretation of the data presented on the figures I strongly suggest to show raw data and mean in case of normal distribution and raw data, median and percentiles in case of skewed distribution.

Response: We have now included raw data in all graphs, and box plots are used in most data representations.

Discussion

I believe this section should be completely rewritten and the strength of conclusions should be reduced after reconsidering the results of the study. The title and abstract should be changed accordingly.

Response: The title and abstract were changed to avoid making statements based on marginal effects (i.e. p -values > 0.05) of the experimental treatments on cannibalistic behaviour. The discussion (pages 9-12) was restructured following indications of both reviewers and the strength of the conclusions was adjusted according to the statistical results.

I am not able to propose major revision of the manuscript in the current form. I hope that my comments will help the Authors to improve the manuscript and I will be very happy to see the improved version published. As an author of several papers I know how hard is the process of publishing a manuscript. While I had to be critical in my evaluation, I keep my fingers crossed for the Authors.

Response: We are grateful for all the comments and suggestions provided and hope that our much revised manuscript is now acceptable for publication.

Reviewer: 2

The authors test the individual and interactive effects of sex, diet nutrient (i.e. protein-to-carbohydrate) balance, and rearing density on cricket food consumption (including eating other crickets); development rate; body mass and composition at maturity; (juvenile?) survival and adult lifespan; and reproduction. The current presentation of the Results precludes the possibility

of clearly stating general outcomes in terms of where there are interactive versus main effects of each tested factor on each type of fitness-relevant outcome. Therefore, it is equally challenging to highlight the broader intellectual insights gained from this study, as might be related to how these factors might exert interactive effects on population dynamics in the wild, where food quality and density almost certainly vary spatially and temporally for many organisms. I recommend extensive revision and reanalysis to ensure the effort that went into this kind of labor-intensive experiment translates into substantive discussion of impactful, clearly communicated outcomes, starting with revision of the Results.

Response: We used the comments and suggestions from both reviewers to restructure the whole manuscript. All sections, and in particular the introduction and discussion, undergone a noticeable change, we are convinced this new version is clearer for the readers.

Restructure the Results to group food consumption and cannibalism results.

Response: Done as suggested. Food consumption and cannibalism are presented consecutively in the results section (page 8, L 32 – page 9, L 7), and Fig. 4 shows the most meaningful results.

How did food consumption rates compare between the two diets? This analysis is missing.

Response: We thank the reviewer for pointing to this. Diet type was included in the statistical model but it was not significant. Therefore, we modified the paragraph to clarify the two points raised by the reviewer in this comment: firstly, we included a sentence at the beginning of the paragraph to state that diet composition did not have a significant effect on the consumption rate (page 8, L 32).

Given that individual females ate more than individual males, there will be differences in variance when comparing food consumption between solitary individuals versus groups. This impacts the statistical comparison and interpretation of the group consumption data.

Response: The increased variance for comparing grouped (males+females) vs solitary crickets was account for by using a GLM model with Gamma distribution. Further details on data analysis are included in a new paragraph in the materials and Methods section (page 5, L 5-15), and model diagnostics are included in the supplementary materials – Fig. S3).

Also to the extent possible distinguish whether the observed cannibalism reflects intraspecific predation versus scavenging. My years of cricket observations suggest the vast majority of incidents are scavenging, aside from the sensitive period right after a cricket molts.

Response: We thank the reviewer for sharing his knowledge on this particular topic. We added a sentence in the discussion section to point to the potential mechanism in which cannibalism may arise (page 10, L 19-20).

Group the body mass and body composition results.

Response: Done as suggested. Individual Body Mass, Protein Content and Lipid Content are presented consecutively in the results section (page 8, L 6-29). Yet, these results are split into Fig. 2 and 3 to keep the visual aesthetics (one or two panels per figure).

Note that Maklakov et al. and Harrison et al. (cited by the authors) have already identified that there are distinct nutritional optima and different nutrient preferences for male vs. female crickets, and thus the sex differences observed here are unsurprising. Also, males expend a considerable amount of energy via singing, a potential explanation for the lipid differences between solitary versus grouped males.

Response: This is a very interesting point concerning nutrients intake and investment. We included several lines in the discussion addressing the potential consequences of increased/decreased lipid content for each sex (page 11, L 13-26).

Be clear about distinctions between measurements of juvenile survival versus adult survival (lifespan).

Response: we modified several lines in the Materials and Methods section to make the separation between juvenile survival and adult lifespan clearer for the readers.

Also, explain why the impacts of grouping on adult female survival and reproduction were not assessed, or provide the analysis and outcomes. The absence of this aspect weakens the manuscript.

Response: We modified this paragraph to state that the effect of the social environment (i.e. grouping) was not statistically significant for both fecundity (page 9, L 11-12) and lifespan (i.e. adult female survival (page 9, L 19-20).

Introduction and Discussion: The current presented intellectual framework is weak. Clear hypotheses, predictions, and mechanistic explanations for outcomes are lacking, and multiple key citations are missing. Novelty alone is insufficient justification. E.g. How, specifically, do the authors expect that nutrient quality will "modulate" the effects of the social environment on insect fitness? I have listed a selection of references at the end of this review that should be incorporated, and would especially recommend consideration of the extensive literature on locusts, where there has been longstanding interest in understanding the interactive effects of diet and density on locust behavior and fitness outcomes.

Response: We restructured the introduction and discussion following the recommendations of both reviewers. Additionally, we re-wrote the hypothesis presented in the final paragraph of the introduction taking into consideration the comments of Reviewer #2 (page 3, L 17-23).

We are as well grateful for the literature suggested, all citations were incorporated both in the introduction and discussion, except those by Hawlena & Schmitz (2010) and Visanuvimol & Bertram (2011) as they did not fit the current storyline (with all the modifications performed).

Discussion point: Crickets could very well be eating other crickets for lipids as an energy source as much as for protein, given that the energy density of lipids is roughly twice as high as the energy density of protein (and the current data illustrate just how lipid-rich the crickets are!).

Response: We thank the reviewer for sharing his thought on this matter. We added a few lines in the discussion addressing this hypothesis (page 10, L 91-33).

Minor comments:

Instead of "batch," use the term "cohort"

Response: replaced in Fig 1, and throughout the Materials and Methods section.

Page 2 lines 23-25: "largely unknown" does not do justice to the substantial body of literature

assessing the physiological mechanisms that regulate nutrient preferences and how animals cope with imbalanced diets (reviewed in depth in reference [54]).

Response: We agree with this comment. Consequently, we deleted this sentence. This paragraph was modified following the comments of both reviewers.

Page 2 lines 50/51: recommend changing "insects" to "animals."

Response: Changed as suggested.

Page 3 lines 28/29: Consider citing Zera and Tiebel (1988), as well as Mole and Zera (1994). References at the end.

Response: Citations included as suggested.

Page 4 line 54: Add an indicator that "grouped" meant 6 crickets per container. Also provide full dimensions of the containers used. Surface area is more relevant than volume for interaction rates.

Response: Modified as suggested (page 3, L34; page 4, L 27).

Page 5 line 37: Which generation(s) of lab-reared crickets were used for the experiments? Consider commentary in Huey and Rosenzweig (2009; especially p. 678 onward) and potential impacts of maternal effects, if the first generation was used.

Response: In this study, we used the first generation of the lab-reared crickets. We included the following sentence to make the readers aware of this potential bias (page 12, L 21-26):

“It's worth noticing that the house cricket *A. domesticus* used in this study may have undergone a process of selection due to artificial rearing conditions during several generations (this species is bred as animal fed [101], and the cohorts used in this experiment came from crickets purchased from a commercial company – see methods for further details). As pointed out by Huey & Rosenzweig [102], the aforementioned likely selection process and the reductional nature of laboratory experiments would prevent us from making generalizations on omnivorous insects in this study, especially regarding wild populations”.

Page 7 line 12: What age were the crickets when homogenized?

Response: Thank you for pointing to this. We standardized the age of collection to 15 days (from adult hatching) but somehow, we did not include this information in the initial version of the manuscript. The modified sentence now reads as follows “Five 15-days-old adult males and females from each treatment (N=40) were individually homogenized in ...” (page 6, L 19).

Page 8 line 21: I assume eggs were counted once a week?

Response: that is correct, we clarify this by modifying this sentence as follows “Egg production was recorded weekly until the natural death of all females ...” (page 7, L 15).

Page 9 line 8: Replace "barely" with "marginally"

Response: replaced as suggested.

Page 11 line 21: You cannot conclude that "individual reproductive output was determined by nutrition alone" if there were no efforts to assess density-dependent effects on reproduction. Clarify this comparison in the Methods and state the outcome in the Results, if it was made.

Response: In our experiment, we subjected the insects to all four experimental treatments since the first instar until adulthood. However, when they reached adulthood, females were individually housed to be able to assess egg production for every female. As the development took considerable time (ca. 60 days on average) we consider that females were exposed to the effects of the social environment (either reared alone or in groups).

To further clarify this for the readers we included the following paragraph (page 11, L 31-35): “Yet, it is worth noticing that the social environment treatment was present only during half of the female lifespan (at least for those being reared in groups). Due to our experiment design, females were housed individually after reaching adulthood to control the frequency of mating and to be able to measure egg production accurately for every individual. Therefore, it is possible that prolonged interspecific interaction may cause a direct effect on female *A. domesticus* reproduction, but we cannot assess such an effect with the data yielded in this study”.

Figure 2, part a, is not particularly informative and could be omitted. I would have expected to see a survival curve.

Response: We removed this graph as suggested.

Appendix C

I have carefully read and thought over the responses presented by the authors to the comments of the reviewers and I have considered the changes made to the manuscript in comparison to its previous version. The authors generally responded to comments by making appropriate changes or by providing a suitable response. The methods and results sections are greatly improved. However, specific predictions and mechanistic explanations/interpretations of main results are still not clear and need further improvement.

I can imagine that testing three factors (sex, diet quality and social environment) on multiple traits might represent a challenge for clear interpretation of obtained results. The main problem is that predictions are not specific (see later my suggestions) and results are to a large extent discussed separately without providing a bigger picture. The studied traits are often correlated (i.e., life-history traits) and therefore cannot be discussed in isolation. As an example, food quality is a strong predictor of age and body size at maturation in ectotherms, and high food quality usually drives fast growth and early maturation (shorter developmental time) at a larger body size (classical life-history trade-off, Berrigan and Charnov 1994). This response can be confounded by the effect of sex giving different selection pressures on males and females (e.g., fecundity selection on females favouring postponement of maturation, reaching larger body size at a later stage compared to males). Another example is the effect of strong intra-specific competition, which can negatively affect juvenile survival, slow juvenile growth and reduce body size at maturation (e.g., Agnew et al. 2002, *Ecol Entomol*). **How the obtained results fit into these complex life-history strategy scenarios?** This is my major comment, I see the potential in the data, but they need to be discussed in more comprehensive way.

What might help is to formulate **specific predictions** with regards the experimental manipulation of factors (current version is still a bit general). There are vast number of studies dealing with the effect of diet quality and intraspecific competition on insect fitness, so authors should not have difficulties to come up with reasonable predictions. Just for an inspiration: If diet quality is important factor for insect life-history traits, we expect shorter/longer development time, higher/lower survival rates, fecundity...etc. If intraspecific competition is important for insect growth and survival, we expect....etc. Specific predictions can be made also with regards to sex differences (this would depend on the strategy of this species, which I unfortunately don't have any knowledge).

It is great that the authors measured other traits such as food consumption, incidence of cannibalism, protein and lipid content. However, how is it connected with the studied questions? For example, food consumption is often used a proxy for assessment of food quality and can be strongly correlated with growth rate. Is it the case for this study? This needs some justification, now it reads a bit random (this again would help the authors to provide more clearer story).

Specific comments

Introduction

Page 1, Lines 14-18: Another classical life-history trade-off is juvenile survival vs adult reproduction, number vs size of offspring... etc (Stearns 1989, *Funct Ecol*)

Page 2, Lines 11-15: What is the life-history strategy of this species? Are females bigger? Do they reach the maturation earlier than males? Are they living in social groups or solitary? Is there strong competition between males? This information would help authors to come up with more specific predictions.

Page 2, Lines 14-24: As I mentioned, specific predictions would help the authors to interpret the results more clearly.

It is also not clear what authors mean by their manipulations. Does protein-rich diet represent high-quality diet? Does manipulation of social environment (increased number of crickets) mimic intra-specific competition? This is important for further interpretation of data and needs further clarification.

Methods

Page 5, Line 4-18: Data analysis paragraph usually comes at the end of the method section

Page 5, Line 21-25: How did authors deal with the case one of the individuals died? Did they add new cricket to the container? This might be important because all the replicates should be kept the same during the whole experimental period (one container with six crickets).

Results

Page 7, Line 25-29: Is there any special reason why author did not include the figure about the effect of social environment on juvenile cricket survival?

Discussion

Page 10 Line 21-33: This paragraph should be discussed later. The flow of discussion should somehow mirror the order of the predictions and the results.

One of the potential explanations of cannibalism is also the need to acquire some symbionts (Brandstädter, K., and Zimmer, M. (2008). "Infection of terrestrial isopods with environmentally transmitted bacterial symbionts," in *Proceedings of the International Symposium on Terrestrial Isopod Biology*; Le Clec'h, W., Chevalier, F. D., Genty, L., Bertaux, J., Bouchon, D., and Sicard, M. (2013a). Cannibalism and predation as paths for horizontal passage of *Wolbachia* between terrestrial isopods. *PLoS ONE* 8:e60232)

Page 11, Line 10-11: If protein-rich diet was high-quality, this explanation doesn't make too much sense. How could excess of protein compensate for late development? If dietary protein was toxic...then we cannot really talk about high-quality diet.

Page 11, Line 27-30: here suddenly nitrogen is not toxic but rather supports high-quality diet manipulation

Page 12, Line 21-26: Hmmm, if using the house cricket from artificial rearing conditions will prevent to make any generalizations on nutritional aspect of this insect....what your results then actually say? I would be careful with such a statement, please rewrite and just mention briefly

Figures

One general note: Females showed to consume more food, and also had higher protein and lipid content. This can be simply explained by their bigger size. Did authors control for the effect of body mass on tested traits in their analyses?

Appendix D

Response to decision letter

Associate Editor Comments to Author (Dr Punidan Jeyasingh):

This revised manuscript was reassessed by the original reviewers. Both reviewers felt the manuscript is much improved. Nevertheless, both of them mention several remaining issues. Again, I felt the reviews were fair and constructive. I invite the authors to make these revisions.

Editor comments:

Thank you for your revisions. As you will see, the reviewers appreciated your improvements but still have concerns. Please be aware that we cannot send your manuscript for review again. If our AE determines that you have not fully addressed all of the remaining concerns in your next version, we will not be able to consider it. Best wishes for your reworking.

***Response:** We are very thankful for having the chance to further improve our manuscript after the second revision round. In this newer version, we did our best to answer all concerns raised by the referees to their full satisfaction and hope that our manuscript now comes in the quality to be accepted for publication in “Royal Society Open Science”.*

Reviewer comments to Author:

Reviewer: 3

Comments to the Author(s)

The paper by Gutiérrez and colleagues looks at the combined effects of social environment and diet composition on life history and physiology of crickets. The paper has already gone through one round of reviews and I can say it is visible, the paper reads well and has logical structure. I especially like the Authors adherence to discussing biological effects instead of purely statistical significances – it’s reassuring seeing papers like this, in the reproducibility era and in the crisis of overly (and wrongly) relied on p-values. However, I have some comments about the general presentation of results, the details of statistical analyses and the interpretation of the whole story.

Statistical analysis

What is the “experimental unit” mentioned in the model description? It’s important as this is critical to establish whether analyses were pseudo-replicated or not. Description of models should be possibly unambiguous. Following comment also makes it very unclear – if females were individually housed in experimental units – they still are not independent (unless the definition of experimental units clears this up for the reader).

Response: We modified L 131 to make clear for the reader that we refer to the plastic container as the experimental unit. Additionally, we changed “container code” to “experimental unit code” in L 149 and 152 to avoid confusion. In some cases, several individuals belonging to the same container were measured (e.g., nymph survival, developmental time and individual body mass). Consequently, mixed-effects models were chosen to analyse these kinds of response variables to properly account for non-independence of observations in these cases.

Specifically, for the example given by the reviewer “females were individually housed in experimental units”, these were indeed independent because every female cricket was confined to a single plastic container. Note that positions of these containers were re-randomized every three days within and between growth cabinets, rendering our design a fully randomized design.

“Integrated by independent replicates” – what does this mean?

Response: This comment is related to the one above. We have now modified this sentence in the L 216-218 to make it clearer: “Generalized linear models (GLM) were used to model most response variables when the replicates were independent (i.e. when only one cricket per experimental unit was sampled).”

Which modelling function (lm? lmer? glm?) as used in R?

Response: This information (i.e. R libraries and functions used) is provided in the statistical analysis section for each response variable. In addition, we now provide a detailed overview of the models in a new table in the Supplementary material (Table S2). Please also see our additional answers to the two subsequent comments related to this.

Also, the section should describe in more detail which model was used for which type of response variable. And how these were decided. In many cases using models such as negative binomial is an overkill, and simpler Poisson models suffice. The same with gamma-distribution models. Was

the decision based only on the characteristics of the data? Or on formal distribution goodness-of-fit tests? Formally testing data for fit to a specific distribution is also not recommended, especially with sample sizes seen here – such tests are overly conservative, and not very informative with smaller sample sizes.

Response: We have now provided a new supplementary table (Table S2) containing this information. Distributional characteristics of all response variables were assessed using the R package “fitdistrplus”, based on graphical interpretation and AIC values. We included this information in L219-222 “). Both types of generalized models (i.e. GLM and GLMM) were fitted using the appropriate link and variance functions (e.g. Gamma, negative binomial), which were assessed using the fitdistrplus library [73]; depending on an assessment of dispersion, we decided among (i) Normal, log-normal or Gamma and (ii) Poisson or negative binomial and (iii) binomial or quasibinomial models [74]”.

[reading on, I see descriptions of models in respective response variables; this is unnecessarily redundant and repetitive; most of the “General data analysis” section could be removed as it is again repeated in subsequent sections]

Response: We here followed a comment by Reviewer 1 in the past revision round, who suggested to expand the “General data analysis” subsection. In the current version of the manuscript, “General data analysis” appears at the end of the materials methods section, and we feel that it correctly informs the reader about the criteria for model selection without detailing the specifics for every model and response variable. We have now added a sentence to make the reader aware that further details on the statistics are given elsewhere in the manuscript: (L222-223): “Further details on the models, variables and functions are available in the Supplementary material (Table S2).” We hope with this and the additional new table, the statistical analysis is now sufficiently clarified.

Why food consumption was analyzed using a gamma-distributed model? I can hardly find a rational justification – a quite likely simple box-cox transformation would render the data manageable by a Gaussian distribution model.

Response: In the analysis of datasets in biology and ecology, the Gamma distribution is generally used for continuous data with more variance than expected under a Normal distribution or data that are skewed to the right. It can be used to analyse almost any positive continuous variable with large variance (concentrations, intensities, growth). A box-cox-transformation would have been an entirely different modelling philosophy (and

would also have transformed the variance, which is not the case in generalized linear models). So, to answer in short, analyzing these data using a Gamma generalized linear model was fully justified. We refer to the book by Benjamin M. Bolker (2008), *Ecological models and data in R*, Princeton University Press, pages 131-132), where the usage of this distribution is discussed in great detail. In summary, our approach was not to transform data but rather to use the appropriate distribution (which was identified by statistical means as explained in the “General data analysis” subsection). Here, models with gamma-distributed errors outperformed others.

Why introduce yet another distribution for egg counts? Log-transformation should work just fine, or (even though it has its limitations) a quasi-Poisson model, or even better – an additive overdispersion model in MCMCglmm.

Response: There are two different modelling philosophies – one is to transform data, the other one is to let this transformation be done “inside” a generalized linear model (by transforming the expected value E). Throughout this manuscript, we adopted the latter approach, i.e. used generalized linear models, as these allow separate modelling of the mean $E(X)$ and the variance. This is a standard practice recommended by statisticians.

To answer the referee’s question, we have now rechecked the data and potential distributions to be used and we concluded that using log-transformation of the response variable would have yielded similar statistical results (in both cases the $P < 0.001$). Therefore, we argue that it’s not necessary to transform data, but rather use the appropriate distribution. The negative-binomial family used for analysing average weekly egg production and total egg production was chosen based on the results from the assessment of data distributions using the library *fitdistrplus* as described in the “General data analysis” subsection (please see figure at the end of this comment showing the graphical output for this).

Additionally, the quasi-poisson distribution is not implemented in the *glmer* function (*lme4* library), which is the function we used for our generalized linear mixed effects models (GLMM). The model we used was very similar to a quasipoisson model. Again, we refer to Bolker, B.M (2008) for an in-depth description of when to use negative binomial distributions.

We did not consider MCMCglmm models here. Instead, we are confident that our analyses are more than appropriate for these data. As we always used a multi-model approach (testing several distributions and models first), we have sufficiently characterized each response variable using a fully reproducible approach. We further

provide all R code and datasets used for analysis so that future analyses with these data are possible.

In cases where you claim overdispersion was tackled using quasi-Poisson – do you have indications the data IS overdispersed?

Response: This comment refers to the statistical analysis performed on the cannibalism rate data. Initially, we concluded that such data was overdispersed through the inspection of the preliminary models formulated with the Poisson distribution, where the residual deviance divided by the residual degrees of freedom was considerably larger than unity. However, after carefully reassessing this model we concluded that quasi-binomial is the adequate distribution to be used given that it is proportional data (it was changed accordingly in L 171). Yet, it is important to note that when the analysis was performed with other distribution families (e.g., Poisson and Negative Binomial), the results were similar, in all cases the effect of diet on cannibalism rate was close to $P=0.5 - 0.6$.

In general, I would encourage providing 95% CIs for all estimates, and not just for some (e.g. for difference in survivorship between groups etc). P values are misleading (as authors correctly argue in the response to reviewers) but this is exactly why CIs are much better in describing biological reality. It may also be using SEs but please be consistent (currently sometimes you use CIs, sometimes SE). Of course, in general confidence intervals are much more informative.

Response: Thank you for pointing to this. We changed all SE (standard errors) to 95% CI (confidence intervals) throughout the results section (L 242-306) following the suggestion of the Reviewer 3.

As for figures: they are much better, but still miss description – e.g. what do box plots describe? Saying “the distribution” is not enough. Are these min-max-IQR values? Are the bands on the hazard plot CIs or SEs? Also – be consistent. You report sometimes values >0.05 , and sometimes you just state n.s. I’m not a fan of p-values, but go for all or none – either everything that is >0.05 is n.s. – or report all p values (of which I’m a bigger fan actually).

Response: The figure captions were modified to include a more detailed description of the box-plots is provided for the captions of Figures 2-5. Additionally, we included a description of the confidence interval for the hazard plot depicted in Figure 6. Finally, we modified Figure 2 to include the P-value instead of “n.s.”.

Discussion

In the discussion you mention that crickets were cannibalizing only external/exoskeletal parts of other crickets (which are made of mostly indigestible chitin – is that right?). This sounds almost as if they were not really eating them but only exhibiting biting/fake eating behavior that is often seen as a response to stress in animals. In that case cannibalism would not be related to food composition and could be just reaction to stress. Are there any studies of other stress sources eliciting similar response? This warrants discussion as it’s an important component of your results.

Response: There is not much information about the exact molecular composition of *Acheta* cuticle. Yet, in *Periplaneta americana*, chitin represents 49% and protein 38% of the content, followed by diphenols (11%) and lipids (2%) (Kramer et al., 1991). Therefore, it is not precise to oversimplify the cuticle composition to mostly indigestible chitin. Besides, Mira (2000) – a study that we already cited in our manuscript – experimentally demonstrated that cockroaches can recycle over 58% of the nitrogen present in the cuticle.

We feel that is unnecessary to extend the discussion of this particular topic (especially regarding stress) as we do not have quantitative data to demonstrate this. Although we observed this behaviour consistently in our experiment, our point of view is not strictly unbiased. We complemented the sentence by including the reference to Kramer et al.,

(1991), the new version reads as follows (L 419-421): “This particular interest in consuming the cuticle of conspecifics could be a homolog to the consumption of exuviae, which are composed mainly of nitrogenous compounds [114], particularly chitin and protein [115].”

Mira A. 2000 Exuviae eating: a nitrogen meal? J. Insect Physiol. 46, 605–610.

Kramer, K. J., Christensen, A. M., Morgan, T. D., Schaefer, J., Czapla, T. H., & Hopkins, T. L. (1991). Analysis of cockroach oothecae and exuviae by solid-state ¹³C-NMR spectroscopy. Insect biochemistry, 21(2), 149-156.

In the discussion you also lend considerable space to the toxic effects of protein excess. However, your paper misses an important detail: what is the composition of diets of wild-type crickets? How different the two treatments you had were from natural conditions?

Response: We address this missing information in the discussion section by adding the following sentences in L 370-373: “It is known that *A. domesticus* is a non-specialized omnivore; in natural conditions, this cricket has been recorded attacking crops, vegetables and feeding on immobile stages of insects (e.g. eggs and pupae) [73]. Yet, the way they mix nutrients and their dietary preferences have only been studied in laboratory conditions [93–96].

In general I like the discussion, it can be seen that it was worked on on structured in a better way.

Lesser points:

(Please use continuous numbering of lines in the future...)

(suggested changes in caps)

Page 2

L10: IN some scenarios

Response: Modified as suggested (L 29).

L11: such as ENVIRONMENTS STRONGLY TRANSFORMED BY ...

Response: Modified as suggested (L 30).

L13: PHENOMENA; EVEN not necessary

Response: Modified as suggested (L 32).

L16: I would remove PLASTICTY; plasticity is a tricky word and usually not equal to “flexible physiology” which may result from simple redundancy/buffering mechanisms.

Response: Removed as suggested (L 34).

L17: materializes – weird wording, maybe “is reflected in” or “manifests”?

Response: We changed the word “plasticity” to “flexibility” according to the past comment, and we changed “materializes” to “manifests” as suggested (L 35).

L21: living AT high DENSITIES

Response: Modified as suggested (L 40).

L29: upper case “Isolation” not necessary

Response: Modified as suggested (L 48).

L32: what makes insects so exceptional? Others animals also eat, and often in much more interesting/biologically complex ways...

Response: We had a more complete paragraph addressing this topic in a previous version of the manuscript. Yet, it was advised to be removed by one of the reviewers. In this new version, we included a couple of sentences to briefly explain the advantages of using insects as model organisms for this kind of studies. Now this section reads as follows:

(L 51-55) “In this study, we use an omnivorous insect species as a model organism as insects are exceptional models for the study of nutritional ecology due to the ease of rearing them and their relatively short lifespan, which allows assessing effects through their lifecycle [49,50]. Besides, insects are ideal systems for the manipulation of the social

environment [51] and there is a considerable body of knowledge on the on optimal nutrient ratios for maximizing fitness for several insect species [52,53].”

Page 3

L31 (and other throughout the text): Don't be afraid to use active voice in scientific papers. Why a long, convoluted sentence that something “was combined” – if you can say “We have ...”?

Response: Modified as suggested. We are grateful for the advice from reviewer 3. We try to use a combination of passive and active voice to make the manuscript more dynamic for the reader.

L31-32: diet quality was manipulated ... through ... diets? Rewrite, e.g. Diet was manipulated by CREATING two isocaloric treatments – or sth similar.

Response: Modified as suggested. Now this sentence reads as follows (L 101-103): “Diet composition was manipulated by creating two isocaloric treatments (i.e., diets) of defined chemical composition (see below) containing either 1:1 or 3:1 protein to carbohydrate ratio”.

L34: solitarily? Perhaps – SOLITARITY?

Response: Modified as suggested (L 104).

Page 6

L15: log function – I'm assuming you are referring to log-LINK function?

Response: The distribution and link for this model were changed as described above. This sentence now reads as follows (L 170-171): “Cannibalism was analysed using GLMs with quasi-binomial family and logit-link function”.

L16: by non-normal errors – you refer to the original distribution of data deviating from normal? I would reword it.

Response: Thank you for pointing to this. We rewrote this sentence because the family distribution for this model was changed to better match the proportional data (see L 170-171).

L14-16: Repetitive, described earlier in the text.

Response: we rewrote this paragraph to avoid duplicated information. This sentences now read as follows (L 201-203): “Egg production was recorded weekly (oviposition substrate similar as described above) until the natural death of all females (ca. 5 months) by washing the eggs from the sand in a 300 µm sieve and counting them under a stereomicroscope.”

L19 may have likely HAPPENED; by the way – do you have evidence that nymphs were more likely to be cannibalized? There is no trace of this in methods/results. If not – leave this out as this is only speculation.

Response: We modified the incorrect word as suggested (L 413). However, we decided to keep this sentence because this is a personal comment shared by Reviewer 2 in the previous revision round as he/she seems to be experienced in working with crickets as well. Additionally, we included a citation (Walker & Masaki, 1989) that gives further support to this statement.

Walker TJ, Masaki S. 1989 Natural history. In “Cricket Behavior and Neurobiology” Ed by F Huber, TE Moore, W Loher.

L33 OF A cricket body – not “from the”

Response: Modified as suggested (L 42).

L5 Drop “then”

Response: Modified as suggested.

I haven't made a thorough editing read of the paper but I suggest – after the whole round of revisions – an error-editing by an English editor, to get rid of possible grammar/lexical mistakes.

Response: We are very thankful for the comments on grammar mistakes. We did our best to correct all typos and grammar issues in this new version of the manuscript.

Reviewer: 4

Comments to the Author(s)

I have carefully read and thought over the responses presented by the authors to the comments of the reviewers and I have considered the changes made to the manuscript in comparison to its previous version. The authors generally responded to comments by making appropriate changes or by providing a suitable response. The methods and results sections are greatly improved. However, specific predictions and mechanistic explanations/interpretations of main results are still not clear and need further improvement.

I can imagine that testing three factors (sex, diet quality and social environment) on multiple traits might represent a challenge for clear interpretation of obtained results. The main problem is that predictions are not specific (see later my suggestions) and results are to a large extent discussed separately without providing a bigger picture. The studied traits are often correlated (i.e., life-history traits) and therefore cannot be discussed in isolation. As an example, food quality is a strong predictor of age and body size at maturation in ectotherms, and high food quality usually drives fast growth and early maturation (shorter developmental time) at a larger body size (classical life-history trade-off, Berrigan and Charnov 1994). This response can be confounded by the effect of sex giving different selection pressures on males and females (e.g., fecundity selection on females favouring postponement of maturation, reaching larger body size at a later stage compared to males). Another example is the effect of strong intra-specific competition, which can negatively affect juvenile survival, slow juvenile growth and reduce body size at maturation (e.g., Agnew et al. 2002, Ecol Entomol). How the obtained results fit into these complex life-history strategy scenarios? This is my major comment, I see the potential in the data, but they need to be discussed in more comprehensive way.

Response: We thank Reviewer 4 for his constructive criticism. We have now added an entirely new analysis section and novel results that help to better understand the life-history trade-offs. In the following comments, we describe the modifications made to the manuscript to better inform the reader about our hypothesis/predictions. We modified the “results” and “discussion” sections as well. See our response to the following comments below, where we explain the details of how we addressed these comments.

What might help is to formulate specific predictions with regards the experimental manipulation of factors (current version is still a bit general). There are vast number of studies dealing with the effect of diet quality and intraspecific competition on insect fitness, so authors should not have difficulties to come up with reasonable predictions. Just for an inspiration: If diet quality is important factor for insect life-history traits, we expect shorter/longer development time, higher/lower survival rates, fecundity....etc. If intraspecific competition is important for insect growth and survival, we expect....etc. Specific predictions can be made also with regards to sex differences (this would depend on the strategy of this species, which I unfortunately don't have any knowledge).

Response: We thank Reviewer 4 for providing these examples that helped us to formulate more comprehensive predictions. The last paragraph of the introduction section (L 86-99) now reads as follows (L 77-94):

“We hypothesized that stressful conditions would negatively affect insect survival, development and reproduction, and that corresponding compensatory mechanisms would be context- and sex-specific due to the different life strategies between males and females. Our specific predictions were as follows: i) immature crickets living in crowded conditions would experience lower survival probabilities due to severe intraspecific competition for resources (e.g. food and shelter), yet, a protein-rich diet could partially alleviate this pressure. ii) for those crickets that managed to reach adulthood, we anticipated that the type of diet they fed upon would determine their phenotype (e.g., development time, size, protein and lipid content), being benefited by a more proteinaceous diet. In this case, females would reach a bigger size in a shorter time, therefore with a higher reproductive potential; and males would benefit as well from a bigger size as this would grant them an advantage in the competition for resources (e.g., food, shelter and partners). However, we expected the social environment to modulate this response, given that a more competitive environment (i.e. intraspecific competition) would interfere with the optimal nutrient level acquisition. iii) we predicted that reproduction would be highly determined by diet constitution as shown by previous studies (e.g. [63]), specifically, female crickets feeding on a protein-rich diet would a high fecundity, at the expense of their lifespan, yet, we expected the social environment to modulate this response as well as the resources gathered during the immature development, and that would be invested into reproduction in a later stage, could be compromised by intraspecific competition. Furthermore, we aimed to elucidate the role of nutrition for natural cannibalism occurring in our model organism”.

It is great that the authors measured other traits such as food consumption, incidence of cannibalism, protein and lipid content. However, how is it connected with the studied questions? For example, food consumption is often used a proxy for assessment of food quality and can be strongly correlated with growth rate. Is it the case for this study? This needs some justification, now it reads a bit random (this again would help the authors to provide more clearer story).

Response: Following the referee's suggestion, we have now included a complementary analysis (structural equation models) integrating the experimental treatments (diet constitution and social environment) and three potentially related response variables: food consumption, development time and body mass. This analysis allowed us to demonstrate the central role of food consumption on the adult cricket phenotype, and we additionally identified a positive covariance between body mass and development time in both sexes. The analysis is described in L 224-236 and the results are presented in L 309-324. We included an additional graph (Fig. 7), and the full statistical results are presented in the supplementary material (Table S3 and S4).

Specific comments

Introduction

Page 1, Lines 14-18: Another classical life-history trade-off is juvenile survival vs adult reproduction, number vs size of offspring...etc (Stearns 1989, Funct Ecol)

Response: Information included as suggested. We thank the Reviewer 4 for suggesting this citation. This sentence now reads as follows (L35-36): "In most cases, such flexibility manifests at the expense of reproductive traits, causing a trade-off between longevity and reproduction [31–33], juvenile survival and adult reproduction, or between number and size of offspring [34]."

Page 2, Lines 11-15: What is the life-history strategy of this species? Are females bigger? Do they reach the maturation earlier than males? Are they living in social groups or solitary? Is there strong competition between males? This information would help authors to come up with more specific predictions.

Response: We extended the information regarding known life-history traits and strategy of the house cricket *A. domesticus* in L 72-76. Now, this paragraph reads as follows: "In natural conditions, the cricket *A. domesticus* occurs in big groups in cultivated areas [57] and it has been suggested that this aggregation behaviour might be mediated by chemical cues [58]. Females are naturally larger than males, and are known to choose the

mating partner based on body size and song acoustic features [59,60]. Males fight over shelter, mating partners and resources (e.g. food) [61,62]”

Page 2, Lines 14-24: As I mentioned, specific predictions would help the authors to interpret the results more clearly. It is also not clear what authors mean by their manipulations. Does protein-rich diet represent high-quality diet? Does manipulation of social environment (increased number of crickets) mimic intra-specific competition? This is important for further interpretation of data and needs further clarification.

Response: We modified the specific predictions to make our experiment clearer for the reader. Additionally, we changed “diet quality” to “diet composition” throughout the manuscript to avoid confusions (this is related with a comment below - Page 11, Line 10-11).

Methods

Page 5, Line 4-18: Data analysis paragraph usually comes at the end of the method section

Response: The “General data analysis” subsection is now presented at the end of the methods section as suggested by Reviewer 4.

Page 5, Line 21-25: How did authors deal with the case one of the individuals died? Did they add new cricket to the container? This might be important because all the replicates should be kept the same during the whole experimental period (one container with six crickets).

Response: We did not replace dead/cannibalised crickets because it was not technically possible. If we would have had to replace every individual, we would have needed to keep many spare crickets (both females and males) from all the four experimental conditions (2 diets * 2 social environments) in identical rearing conditions. Besides, we didn't want to include potential confounding factors (e.g., cricket age) by adding individuals into the containers as we aimed to track the long-term effects of the combined experimental factors.

We have clarified this in the manuscript to make the methods clearer for the reader; the sentence in L 169-170 now reads as follows “Every dead specimen was thoroughly inspected for signs of consumption and scored as dead or cannibalized. Dead crickets were removed from the container without replacement”.

Results

Page 7, Line 25-29: Is there any special reason why author did not include the figure about the effect of social environment on juvenile cricket survival?

Response: Juvenile cricket survival was analysed using a mixed-effects Cox model, and the graphical output of such a model is a forest plot (see below, this figure had been included in the first version of the manuscript). It was suggested by Reviewer 2 in the previous revision round that this plot was not very informative, consequently, it had been removed. We included (in the past revision) detailed results on the effects of the experimental treatments on juvenile cricket mortality (L 243-246): “The social environment was the only factor that had a noticeable effect on house cricket survival. Insects living in groups had an increased hazard of death compared to those living solitarily ($X^2 = 9.68$, $P = 0.002$), yet, immature crickets feeding on diets of composition did not have their survival probability affected. Living in groups increased the hazard by a factor of 0.65 (0.49 - 0.85, 95% CI)”. We decided to not include this Figure at this stage, but we would be happy to do so if the Editor thinks this is necessary.

Discussion

Page 10 Line 21-33: This paragraph should be discussed later. The flow of discussion should somehow mirror the order of the predictions and the results. One of the potential explanations of cannibalism is also the need to acquire some symbionts

(Brandstädter, K., and Zimmer, M. (2008). “Infection of terrestrial isopods with environmentally transmitted bacterial symbionts,” in *Proceedings of the International Symposium on Terrestrial Isopod Biology*;

Le Clec'h, W., Chevalier, F. D., Genty, L., Bertaux, J., Bouchon, D., and Sicard, M. (2013a). Cannibalism and predation as paths for horizontal passage of *Wolbachia* between terrestrial isopods. *PLoS ONE* 8:e60232)

Response: We thank the Reviewer 2 for suggesting these interesting publications. This paragraph was expanded by including this sentence at the end: “An additional, non-exclusive, motivation of cannibalistic behaviour could be the need to acquire microbial symbionts [119,120], but this remains to be studied in the cricket *A. domesticus*.” Additionally, we repositioned this paragraph at the end of the discussion (L 426-429).

Page 11, Line 10-11: If protein-rich diet was high-quality, this explanation doesn't make too much sense. How could excess of protein compensate for late development? If dietary protein was toxic...then we cannot really talk about high-quality diet.

Response: We use “diet composition” instead of “diet quality” throughout the text (including the title) to avoid misleading statements. Additionally, we consistently refer to diets as balanced (1:1) and protein-rich (3:1) regarding their Protein : Carbohydrates ratio.

Page 11, Line 27-30: here suddenly nitrogen is not toxic but rather supports high-quality diet Manipulation

Response: This is related to the answer immediately above, we changed and unified the way to describe the diets used in our study (as “high-quality” would certainly have been misleading).

Page 12, Line 21-26: Hmmm, if using the house cricket from artificial rearing conditions will prevent to make any generalizations on nutritional aspect of this insect....what your results then actually say? I would be careful with such a statement, please rewrite and just mention briefly

Response: While we are conscious that all laboratory animals are subject to evolutionary changes, we agree with the reviewer in that this sentence had to be rewritten. The new version reads as follows (L 438-443): “It's worth noticing that the house cricket *A. domesticus* used in this study may have undergone a process of selection due to artificial rearing conditions during several generations (this species is bred as animal fed [123], and the cohorts used in this experiment came from crickets purchased from a commercial company – see methods for further details). As pointed out by Huey & Rosenzweig [124], the aforementioned likely selection process and the reductional nature of laboratory experiments would hinder the generalization of the results obtained after this kind of studies to wild populations.”

Figures

One general note: Females showed to consume more food, and also had higher protein and lipid content. This can be simply explained by their bigger size. Did authors control for the effect of body mass on tested traits in their analyses?

Response: In the newly included structural equation model, we demonstrate that body mass (in both sexes) was influenced by food consumption. Additionally, body mass showed a positive covariance with development time.

Specifically for protein and lipid content, we analysed both total and relative contents of the macromolecules contained in the cricket body.